

# Combined Impacts of Uncertainty in Precipitation and Air Temperature on Simulated Mountain System Recharge from an Integrated Hydrologic Model

Adam P. Schreiner-McGraw[1], Hoori Ajami[1]

[1]Department of Environmental Sciences, University of California, Riverside, 92521, USA.

*Correspondence to*: Adam P. Schreiner-McGraw (adampschreiner@gmail.com)

**Abstract.** Mountainous regions act as the water towers of the world by producing streamflow and groundwater recharge, a function that is particularly important in semiarid regions. Quantifying rates of mountain system recharge is difficult, and hydrologic models offer a method to estimate recharge over large scales. These recharge estimates are prone to uncertainty
from various sources including model structure and parameters. The quality of meteorological forcing datasets, particularly in mountainous regions, is a large source of uncertainty that is often neglected in groundwater investigations. In this contribution, we quantify the impact of uncertainty in both precipitation and air temperature forcing datasets on the simulated groundwater recharge in the mountainous watershed of the Kaweah River in California, USA. We make use of the integrated surface water – groundwater model, ParFlow.CLM and several gridded datasets commonly used in hydrologic studies, downscaled NLDAS-
2, PRISM, Daymet, Gridmet, and TopoWx. Simulations indicate that across all forcing datasets, mountain front recharge is an important component of the water budget in the mountainous watershed accounting for 25 – 46% of the annual precipitation, and ~90% of the total mountain system recharge to the adjacent Central Valley aquifer. The uncertainty in gridded air temperature or precipitation datasets, when assessed individually, results in similar ranges of uncertainty in the simulated water budget. Variations in simulated recharge to changes in precipitation (elasticities) and air temperature (sensitivities) are larger
than 1% change in recharge per 1% change in precipitation or 1-degree C change in temperature. The total volume of snowmelt is the primary factor creating the high water budget sensitivity; and snowmelt volume is influenced by both precipitation and air temperature forcings. The combined effect of uncertainty in air temperature and precipitation on recharge is additive, and results in uncertainty levels roughly equal to the sum of the individual uncertainties. Mountain system recharge pathways including mountain block recharge, mountain aquifer recharge, and mountain front recharge are less sensitive to changes in
air temperature than changes in precipitation. Mountain front and mountain block recharge are more sensitive to changes in precipitation than other recharge pathways. The magnitude of uncertainty in the simulated water budget reflects the importance of developing high qualify meteorological forcing datasets in mountainous regions.



# 1 Introduction

Mountainous catchments are known to be important sources of water in semiarid and seasonally dry ecosystems
(Viviroli et al., 2007). While it is well understood that mountain systems provide the majority of freshwater resources via
streamflow (Viviroli and Weingartner, 2004), the contribution of mountain systems to groundwater resources remains highly
uncertain (Ajami et al., 2011). As meteorological conditions are the primary drivers of the hydrologic cycle, understanding
how groundwater recharge in mountain systems reacts to different meteorological forcings is important. Since mountain
recharge processes have been defined in various ways, we define three distinct recharge pathways in mountain catchments.
Mountain bedrock aquifer recharge (MAR) consists of snowmelt or rainfall derived infiltration into the bedrock system of the
mountain block, which either discharges to streams or may eventually reach an alluvial aquifer in an adjacent valley as
mountain block recharge (MBR). MBR consists of lateral subsurface flow from the mountains to an adjacent valley aquifer.
Finally, mountain front recharge (MFR) consists of direct infiltration of streamflow, that originated in the mountains, along
the piedmont zone. Various efforts have been conducted to estimate the relative importance of each recharge pathway (Ajami
et al., 2011; Mailloux et al., 1999; Manning and Solomon, 2003; Newman et al., 2006; Schreiner-McGraw and Vivoni, 2017),
but an analysis of how they respond to uncertainty in atmospheric drivers, such as precipitation or air temperature, is lacking.

Hydrologic models are important tools to quantify recharge rate as a function of precipitation because recharge rates
are difficult to measure, especially over large spatial extents (Scanlon et al., 2002). Physically based, integrated hydrologic
models that simulate land surface – subsurface hydrologic processes have high computational requirements, but are the best
modeling tools to study connections between meteorological variability and hydrologic function. Furthermore, they are not
limited to empirical relationships or calibrated parameters to a set of historical conditions (Fatichi et al., 2016). Hydrologic
models, however, are prone to uncertainty that can arise from many sources including the model structure, the selection of
equations to represent processes, parameterization, and uncertainty in the model forcing data (Beven, 2006; Woldemeskel et
al., 2012). The impact of the uncertainty in forcing data upon model performance is particularly important when models are
used to assess the impact of climate change or drought on groundwater processes.

The hydrologic system response to changes in precipitation and air temperature has been studied in depth, the impact
of meteorological changes on groundwater, however, has received comparably less attention. It has been shown that the
physiographic features of a watershed, particularly those that control the depth to the water table (DTWT), impact the
groundwater system response to climate variability, but the depth at which these sensitivities are highest is highly uncertain.
Some authors suggest higher groundwater sensitivity to meteorological variability at regions with high DTWT, while others
find higher sensitivity for shallow water table regions (Erler et al., 2019; Maxwell and Kollet, 2008). In a recent review, the
direct impacts of climate on groundwater is explained by describing processes that control the water surplus (precipitation –
evaporation). While precipitation and air temperature impact the magnitude of water surplus, subsurface geology controls the
translation of water surplus (potential recharge) to groundwater head variability (Taylor et al., 2013). The precise impacts of
meteorological variability on groundwater recharge, particularly in mountainous catchments that supply the majority of water



in semiarid regions, remain important unknowns (Meixner et al., 2016). Several studies have used hydrologic models to examine how meteorological focings impact mountain recharge processes, but none has considered the importance of meteorological forcing uncertainty on recharge estimates (Ajami et al., 2012; Crosbie et al., 2011; Hartmann et al., 2017; Schreiner-Mcgraw et al., 2019). This is particularly important in mountainous regions where observational datasets (e.g.,

forcings, subsurface structure, and parameters) are scarce.

The water budgets in mountainous watersheds are typically dominated by snow processes. As a result, the two most important meteorological variables for controlling the hydrologic response are precipitation amount and air temperature. Datasets of both variables are highly uncertain, particularly in regions with high relief, and it is difficult to determine which variable is more uncertain as they have different units (Daly et al., 2008; Henn et al., 2018; Lundquist et al., 2015). From a

hydrologic standpoint, the more important question is whether the level of uncertainty contained in precipitation or air temperature has larger impacts on the simulated hydrologic budget. Recent work in the Colorado River basin has demonstrated the importance of air temperature to simulated hydrologic processes, particularly in regions with snow (Udall and Overpeck, 2017). Climate change is expected to alter both precipitation and air temperature, but their relative changes are unknown, especially for precipitation. It is therefore important to understand how air temperature and precipitation uncertainty might

combine, over a range of conditions, to impact simulated subsurface hydrologic response.

Gridded precipitation and air temperature datasets are especially uncertain in mountainous regions due to a lack of gauges and sharp topographic gradients that alter meteorological conditions over relatively small scales. Previous efforts to test the accuracy of gridded precipitation datasets in mountainous regions have found that datasets are particularly uncertain at the highest elevations (Henn et al., 2018; Lundquist et al., 2015). These uncertainties have been attributed to poor

representation of snow (Rasmussen et al., 2012) and the lack of gauges due to poor infrastructure (Lundquist et al., 2003). The lack of gauges requires extrapolation of meteorological values from gauges in different locations. Gridded datasets vary in their extrapolation techniques of gauge based observations, their use of different input gauges, and their consideration of snow measurements (Daly et al., 1994; Thornton et al., 1997). As a result, there is considerable uncertainty in both precipitation and air temperature gridded datasets that has the potential to alter hydrologic simulations.

In this study, we utilize an integrated surface water-groundwater hydrologic model to study the propagation of uncertainty in precipitation and air temperature into the groundwater system of a mountainous watershed. The model domain encompasses the Kaweah River watershed in California, USA. This domain covers a wide range of climate and topographic conditions and is prone to high inter-annual variability in climate conditions and strong prevalence of drought. We focus on understanding the physical properties that affect the propagation of uncertainty from the atmosphere to the groundwater, and

our discussion aims to answer the three following questions. (1) Which mountain recharge pathway is most impacted by meteorological uncertainty? (2) Is uncertainty in precipitation or air temperature forcing more impactful on the simulated water budget of a mountain system, especially with regards to groundwater processes? (3) How does uncertainty in precipitation combine with uncertainty in air temperature to impact simulated groundwater recharge?





## 2 Methods

### 2.1 Study Site

Model simulations are carried out in the Kaweah River watershed, located in the southern Sierra Nevada Mountains in California, USA (Fig. 1). This location was selected for the study because of the presence of large topographic gradients (elevation ranges from 57 to 4,354 m), steep slopes, and locations with both high and low uncertainty in air temperature and precipitation datasets (Schreiner-McGraw and Ajami, 2020). We identify the Kaweah Terminus sub-watershed, which encompasses the mountainous portion of the Kaweah River watershed upstream of the Terminus dam to investigate the mountain system recharge processes. Furthermore, this undisturbed portion of the domain makes streamflow validation possible. In the Kaweah River watershed, the regional topography is dominated by the Sierra Nevada mountain block, which is largely composed of granitic rocks (Jennings, 1977). The eastern Sierra Nevada mountains contain the tallest peaks in the continental United States and are located in the eastern portion of the study domain. A complex assemblage of landforms composes the piedmont slope of sediments eroding off of the western portion of the mountain range, where our study is focused (Olmsted and Davis, 1961). The elevation decreases to the west of the study domain until reaching the flat Central Valley province. The Central Valley province (Fig. 1) is composed of interbedded sand and silt layers and is a highly productive groundwater aquifer (Faunt, 2009). The climate in the region is a Mediterranean climate with cool, wet winter seasons and hot dry summers. The precipitation in the study domain ranges from ~140 mm – 1,400 mm per year roughly following the elevation gradient. As a result, the vegetation also ranges from desert grasslands (and irrigated agriculture) in the lowlands to oak savannahs and pine forest in the mountain regions.

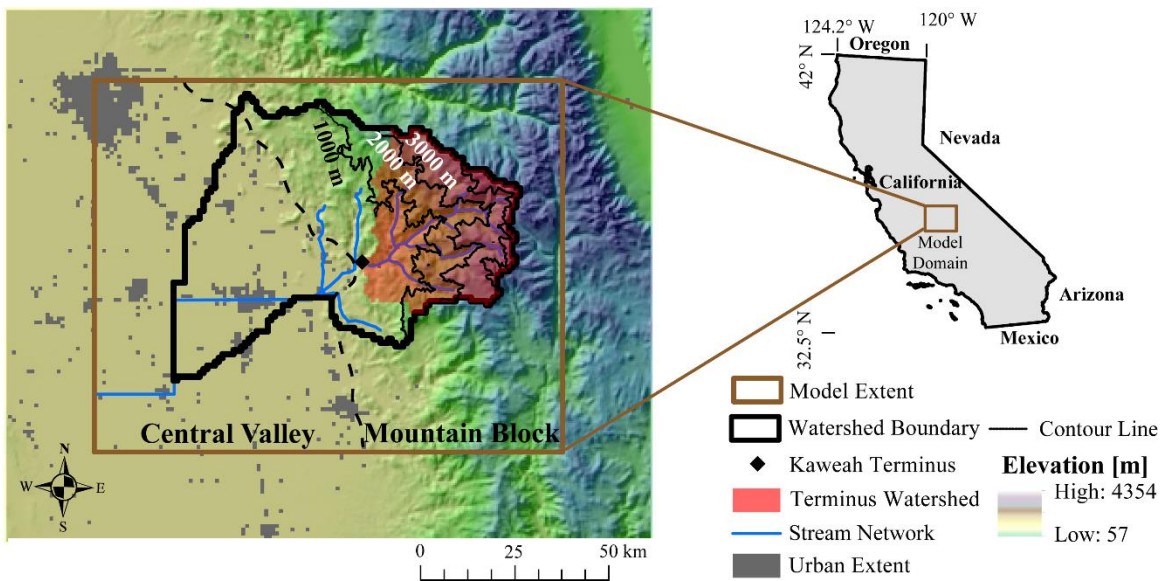

**Figure 1: The location of the model domain within the state of California, USA. A 30-m digital elevation model is used to delineate the Kaweah Terminus watershed and Kaweah River watershed boundaries. The model extent is larger than the watershed boundary**



**to reduce the impact of boundary conditions on simulated groundwater flow. The dashed line indicates the boundary between mountain block and Central Valley aquifer system defined by using a geologic map of the region.**

### 2.2 Model Description

In this study, we use the ParFlow.CLM integrated hydrologic model code (Kollet and Maxwell, 2006; Maxwell, 2013;
Maxwell and Miller, 2005) for hydrologic simulations. The ParFlow.CLM model simulates variably saturated subsurface flow that is fully integrated with overland flow and is coupled to the land surface model CLM 3.0 (Dai et al., 2003). The ParFlow model solves the Richards' equation in three dimensions to simulate variably saturated subsurface flow and simultaneously solves the kinematic wave approximation to simulate overland flow. Channel networks are not predefined in the model, rather they develop naturally in response to the hydrologic conditions and the uniform application of the kinematic wave
approximation to every cell in the model domain. ParFlow has been coupled with the Common Land Model 3.0 (Dai et al., 2003) to simulate the land surface water and energy budgets. The CLM portion of the code interacts with ParFlow over the top soil layers where ParFlow simulates water movement and feeds the soil water state into CLM. We apply the terrain following grid formulation of ParFlow that is best suited to simulate domains with high topographic relief (Maxwell, 2013).

Prior efforts parameterized the model using estimates of topography, land cover type, drill core data, and geologic
maps of the study region (Schreiner-McGraw and Ajami, 2020). A detailed description of the model construction and validation can be found in Schreiner-McGraw and Ajami (2020). Here, we present the conceptual framework relevant to this study. We conceptualize the study domain in two primary physiographic regions, the Sierra Nevada mountain block and the Central Valley, which contains a highly productive aquifer. We apply a 1 km horizontal grid resolution to the 12,276 km$^2$ study domain resulting in a horizontal model grid of 99 x 124. We focus on the groundwater system that is likely to interact with the surface
water and therefore simulate the domain to a depth of 622 m. This depth is consistent with a conceptual model that includes 2 m thick surface soils consisting of 6 layers (0.05, 0.1, 0.15, 0.3, 0.4, and 1.0 m thick) that overlay a 620 m thick aquifer system consistent with observations from drill cores (Faunt, 2009). Surface soil parameters including the saturated hydraulic conductivity, porosity, and van Genuchten parameters are derived from the POLARIS dataset (Chaney et al., 2016). The alluvial aquifer of the Central Valley is conceptualized as 9 rock layers of variable thickness and parameterized following drill
core data compiled by Faunt (2009). The mountain block subsurface is conceptualized as a fractured bedrock aquifer system with three geological layers, saprolite (15 m thick), fractured bedrock (145 m thick), and less fractured bedrock (460 m thick). The mountain bedrock is characterized by low porosity and hydraulic conductivity values that are derived from a geologic map and reference tables (Jennings, 1977; Welch and Allen, 2014). The land surface requires Manning's n values and slope values. Manning's n parameters are based on reference table values (Chow, 2009) and slopes are derived from a 30 m digital
elevation model obtained from the National Elevation Dataset (Gesch et al., 2018). Vegetation types are based on the USDA CropScape data and are aggregated to the IGBP classification system.

The hydrologic model is run at an hourly time step over the water year (WY) 2016 simulation period. We chose WY2016 because remote sensing products were available for model validation, and the meteorological conditions were




approximately representative of the average conditions in the study watershed. The hourly meteorological datasets required as
model forcing include precipitation, air temperature, air pressure, specific humidity, downward short and long wave radiation,
and wind speed in the x and y directions. We obtain all meteorological forcings, except precipitation (*P*) and air temperature
(*TA*), from the Princeton CONUS Forcing dataset, which provides hourly forcings at 3-km spatial resolution based on the
NLDAS-2 dataset. This dataset downscales the NLDAS-2 precipitation dataset using Stage IV and Stage II radar products
(Beck et al., 2019; Pan et al., 2016). Additional precipitation and air temperature forcings are derived from several publically
available gridded datasets; Daymet (Thornton et al., 1997), Gridmet (Abatzoglou, 2013), PRISM (Daly et al., 1994), and
TopoWx which only includes daily minimum and maximum air temperature (Oyler et al., 2015) (Fig. 2). The Daymet, Gridmet,
and PRISM datasets provide daily total precipitation as well as the daily minimum and maximum temperature. These daily
precipitation datasets are downscaled to hourly resolution by applying the temporal downscaling method of NLDAS-2
precipitation.

160         Model initialization consists of a two-step spin up process to bring the subsurface water storage into dynamic
equilibrium with the meteorological conditions. In the first step of the initialization, we start from a initially dry system and
run the ParFlow code without CLM by applying a constant in time net precipitation flux (*P-ET*) (Livneh et al., 2013) to fill up
the groundwater storage and create a rough approximation of the flow network. From this point, each model scenario is run
recursively using the ParFlow.CLM code and the WY2016 forcing data applied in that scenario (see scenario descriptions in
section 2.3). Recursive simulations are continued until the total subsurface storage reaches dynamic equilibrium (Ajami et al.,
2014). We define dynamic equilibrium as the point in which the absolute change in total subsurface storage becomes less than
0.01% in recursive simulations (Ajami et al., 2015).

          Model performance is extensively validated in Schreiner-McGraw and Ajami (2020). As we are focused on
quantifying the impact of air temperature, we present a limited validation primarily related to the energy budget. An important
component of the land surface energy balance in mountainous terrain is the role of snow. We validate model performance
using a reanalysis gridded product that contains estimates of snow water equivalent (*SWE*) and snow covered area (*SCA*) for
the majority of the Sierra Nevada (Margulis et al., 2016). This 90 m resolution dataset is generated using a Bayesian data
assimilation technique with remotely sensed estimates of snow covered area (Margulis et al., 2016). The dataset is clipped to
1,500 m elevation to remove uncertainty related to the infrequent snow below this elevation. When making comparisons
between this reanalysis dataset and our simulated datasets, we also set *SWE* and *SCA* below 1,500 m elevation to 0.
Additionally, we use remote sensing estimates of evapotranspiration (MOD16A2 product) at 1 km resolution from the MODIS
Terra satellite to compare with simulated evapotranspiration (*ET*) and test performance of the simulated energy budget.





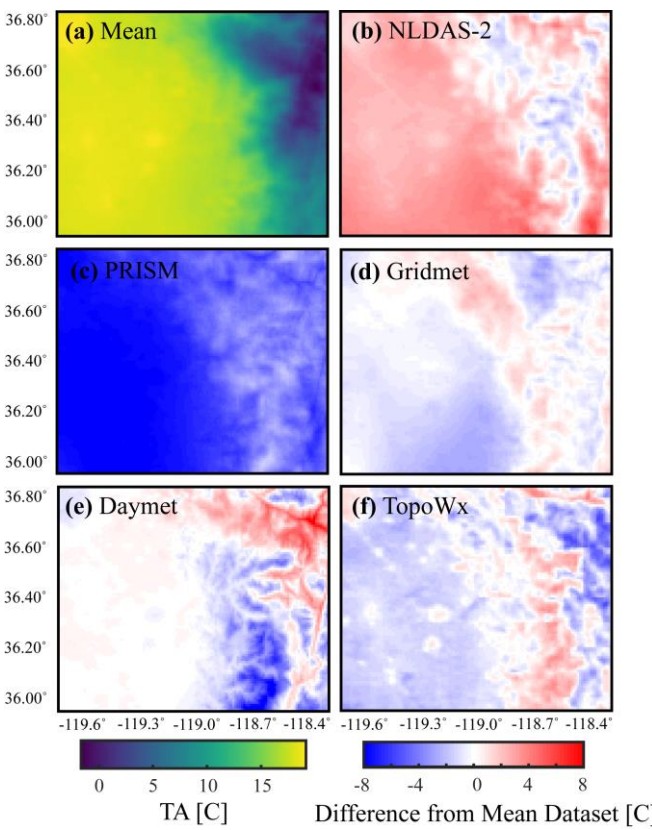

**Figure 2:(a) Mean daily air temperature from 5 air temperature datasets used within this study. Spatial maps represent differences in mean annual daily temperature from the mean dataset (calculated as: dataset – mean) in (b) downscaled NLDAS-2, (c) PRISM, (d) Gridmet, (e) Daymet and (f) TopoWx forcing datasets.**

## 2.3 Model Experiments

In this study, we are interested in quantifying how the uncertainty in air temperature and precipitation focings impact
the simulated water budget. To simplify the system and reduce the impact of uncertainty in anthropogenic management practices, we treat the system as a quasi-pre-development state that is not impacted by groundwater pumping, irrigation, or stream diversions. As a result, all of our model scenarios use consistent parameterizations for the subsurface and the land surface, and the only difference is in the air temperature and precipitation forcings from different gridded meteorological products. We perform a 'base case' simulation where we use the mean precipitation from the 4 available datasets (Daymet,
Gridmet, downscaled NLDAS-2, and PRISM), and the mean air temperature from the same four datasets plus the TopoWx dataset. Prior efforts have demonstrated that using the mean of the precipitation datasets results in the best model performance compared to simulations with each product individually (Schreiner-McGraw and Ajami, 2020). This base case scenario is used for comparison purposes. In addition to the base case scenario, we run three different numerical experiments: (1) variable precipitation and constant air temperature (VarPConstTA), (2) constant precipitation and variable air temperature



(ConstPVarTA), and (3) variable precipitation and variable air temperature (VarPVarTA). In experiment 1, VarPConstTA, we run four scenarios each using the mean air temperature and one of the four precipitation datasets. Experiment 2 is the opposite with 5 scenarios, where each scenario is forced with the mean precipitation and one of the five air temperature datasets. Finally, experiment 3 consists of four scenarios and each scenario is forced with the precipitation and air temperature from one of the four available gridded products.

**2.4 Analysis Techniques**

**2.4.1 Relative Importance of Uncertainty in Precipitation and Air Temperature on the Simulated Water Budget**

We first assess the uncertainty in the precipitation and air temperature datasets by calculating the mean absolute difference (MAD) between each pair of datasets at a daily scale for each grid cell in the domain (Henn et al., 2018). We calculate the MAD between a pair of datasets at a single grid cell as

$$MAD_{i,j} = \frac{1}{d}\sum_{k=1}^{d} \mathrm{abs}(P_{i,k} - P_{j,k}) \hspace{3cm} (1)$$

where $i,j$ represents the difference between dataset $i$ and dataset $j$, $k$ represents the day, and $d$ is the number of days in the year. We calculate the MAD for each pair of datasets and take the mean value of all MADs to represent the mean uncertainty in total precipitation. The same approach is applied to air temperature. We acknowledge that this is not a true measure of uncertainty in precipitation or air temperature as ground truth data from weather stations are not available.

Next, we assess the relative importance of uncertainty in the precipitation and air temperature forcing datasets on the annual water budget partitioning from each simulation scenario. We perform this calculation for the Kaweah Terminus watershed, upstream of the Terminus dam, (Fig. 1) to focus on the mountain groundwater system. The Terminus dam is not represented in the model, and streamflow evaluation downstream of this point is difficult. We calculate the groundwater flux ($GW$) out of the Kaweah Terminus watershed as a residual of the annual water balance, $GW = P - ET - Q - dS$, where $P$ is the

precipitation, $ET$ is the evapotranspiration, $Q$ is the streamflow, and $dS$ is the change in subsurface storage. This groundwater flux is equivalent to the mountain block recharge ($MBR$) that is generated within the Kaweah Terminus watershed. We additionally calculate the precipitation partitioning into rain and snow components. The version of CLM in the model uses a threshold air temperature of 2.5 °C to partition precipitation, so we apply the same threshold to the precipitation data to determine snowfall and rainfall.

Given the seasonality of the water balance in the study watershed, we also calculate the monthly relative range of hydrologic fluxes from the Kaweah Terminus watershed to determine months with the highest uncertainty in simulated fluxes. The relative range ($R_r$) is defined as the range in monthly simulated hydrologic fluxes for each experiment divided by the monthly value from the base case scenario.

**2.4.2 Relative Elasticity and Sensitivity Metrics to Changes in Precipitation and Air Temperature**





To determine the relative sensitivity of the simulated annual hydrologic budget to precipitation and air temperature forcings, we calculate the sensitivity and elasticity of multiple hydrologic variables relative to the baseline simulation for the Kaweah river watershed (Fig. 1). We perform these calculations using the catchment averaged values from experiment 1 ($P$ elasticity) and experiment 2 ($TA$ sensitivity) simulations where $P$ and $TA$ are modified individually. The precipitation elasticity

($\varepsilon$) is the fractional change in a hydrologic variable $v$ from dataset $i$ divided by the fractional change in $P$ from dataset $i$, both relative to our base case scenario.

$$\varepsilon = \frac{\frac{v_i - v_{base}}{v_{base}}}{\frac{P_i - P_{base}}{P_{base}}},$$    (2)

Following the reasoning from Vano et al. (2012), we also calculate the temperature sensitivity ($S$) in a similar manner. We define $S$ as the percent change of a hydrologic variable $v$, caused by a change in $TA$.

$$S = \frac{\frac{v_i - v_{base}}{v_{base}}}{TA_i - TA_{base}},$$    (3)

While we cannot directly compare whether $TA$ or $P$ uncertainty adds more variability to hydrologic simulations, by comparing the $\varepsilon$ and $S$ we can determine whether the range of uncertainty in $TA$ or $P$ contained in common gridded datasets adds more uncertainty to the simulated hydrologic budget. We recognize that the $\varepsilon$ and $S$ are overestimated in this analysis because the datasets have different spatial patterns in $TA$ and $P$, and the basin average differences in simulated hydrology are not solely

caused by the basin average differences in $TA$ and $P$. We contend, however, that this is a reliable approach to estimate the relative importance of model forcing dataset selection. We also assess spatial variability of precipitation elasticities and temperature sensitives by applying Equations 2 and 3 at pixel scale.

### 2.4.3 Impact of Combined Uncertainty in Precipitation and Air Temperature on the Simulated Water Budget

As a result of climate change, both $P$ and $TA$ are expected to change simultaneously. In the analysis described above,

we only alter $P$ or $TA$ individually in experiments 1 and 2, respectively. We make use of the scenarios from experiments 1, 2, and 3 to examine the combination effects of uncertainty in both $P$ and $TA$ on simulated hydrologic response in the Kaweah River watershed. We calculate the relative change in a hydrologic variable, $v$, relative to our 'base case' scenario forced with the mean of both the air temperature and precipitation datasets. For each forcing dataset (Daymet, Gridmet, etc.), we calculate the individual relative difference in simulated hydrologic fluxes or states caused by changing the precipitation dataset ($v_{\Delta P}$)

and the temperature dataset ($v_{\Delta TA}$) from the base case using catchment averaged values. We then estimate the total relative differences in simulated hydrology caused by the combined changes in $P$ and $TA$ by summing the relative differences of $P$ and $TA$ as if there were no interaction effects ($v_{\Delta P \Delta TAest}$) (Vano et al., 2012)

$$V_{\Delta P \Delta TAest} = \frac{(V_{\Delta P} - V_{base})}{V_{base}} + \frac{(V_{\Delta TA} - V_{base})}{V_{base}}$$    (4)





The estimated combined impact of *P* and *TA* changes on the variable, *v*, are then compared to the simulated values of a given
variable when both *P* and *TA* are simultaneously altered in model simulations ($v_{ΔPΔTA}$) to determine the degree of interaction
effects for both variables in the Kaweah River watershed.

### 2.4.4 Sensitivity of Recharge Pathways to Meteorological Forcings

We make use of the integrated hydrologic model to examine the sensitivity of different recharge pathways to changes
in *P* and *TA* forcing. We calculate recharge via three primary pathways, *MAR* (derived from rain or snow), *MBR*, and *MFR*.
We calculate each of these fluxes using the simulated pressure head and saturation values and the Richards' Equation (Maxwell
and Miller, 2005) for specific regions of the model domain. *MAR* is defined as the vertical flux of water leaving the 2 m deep
soil zone (potential recharge) within the Kaweah Terminus watershed, located upstream of the Terminus dam in the Sierra
Nevada Mountains (Fig. 1). We separate *MAR* derived from snowmelt as *MAR* that occurs in the same model time step that
snowmelt occurs (i.e. changes in daily *SWE* is negative), otherwise we assume that *MAR* is sourced from rainfall. We estimate
the *MBR* sourced from the mountainous region of the Kaweah Terminus watershed as a residual of the water balance that is
equivalent to the *GW* flux out of the watershed. We recognize that this is not explicitly *MBR* because the Kaweah Terminus
boundary does not exactly trace the boundary between the mountain block and the valley aquifer. However, the regional flow
pathways ensure that groundwater leaving the Terminus watershed will reach the Central Valley aquifer. Finally, *MFR* is
calculated as the volume of streamflow that infiltrates into the channel bottom as the Kaweah River flows across the piedmont
slope, defined as the area adjacent to the mountain block where topographic slope is greater than 2% (11 km of the Kaweah
River reach).

Previous efforts have shown the role of topography in the propagation of uncertainty in precipitation to groundwater
(Schreiner-McGraw and Ajami, 2020). To examine how this propagation impacts *MAR* under the combined *P* and *TA*
uncertainty versus individual uncertainty in *P* or *TA*, we make use of the relationship between topographic wetness index (*TWI*)
and uncertainty in simulated *MAR* where the *TWI* is calculated as:

$$TWI = \ln\left(\frac{A_C}{\tan \alpha}\right), \tag{5}$$

where $A_C$ is the contributing drainage area and *α* is the slope (Beven and Kirkby, 1979). As the *TWI* is meant to be applied in
climatically similar regions, we apply the analysis only to the Kaweah Terminus watershed where land cover and subsurface
geology are constant, and climate is relatively similar (mean annual precipitation ranges from 435 to 960 mm/yr and mean
annual *TA* ranges from 0 to 15 °C). We estimate the uncertainty in the simulated *MAR* as the standard deviation of *MAR* values
from the multiple scenarios in each *TWI* bin.


## 3 Results and Discussion

### 3.1 Air Temperature and Precipitation Uncertainty

Differences in mean annual daily temperature from the mean temperature dataset range between -8 to 8 ⁰C (Fig. 2b-
f). We analyze the uncertainty in the forcing datasets by presenting the average MAD between the datasets available for *TA*
(Fig. 3a) and *P* (Fig. 3b). Figure 3c presents the annual mean daily MAD averaged across the 5 temperature datasets. Overall,
the uncertainty in air temperature is high with large portions of the model domain expressing an average MAD greater than 5
°C/day. MAD in the topographically flat portion of the domain in the Central Valley is relatively consistent with values of
approximately 5 °C/day. The mountainous region of the study domain has more variability in temperature-based MAD
estimates. Coincidentally, the majority of the mountainous portion of the Kaweah River watershed has relatively low MAD in
*TA* and mountainous regions outside the watershed boundary have much higher uncertainty in *TA* that in places exceeds 7
°C/day. Uncertainty in *P* follows a more consistent pattern than uncertainty in *TA* where the MAD in *P* increases consistently
with elevation (Fig. 3d). This pattern is partially attributable to the annual total precipitation increases in the high elevation
regions, but the lack of meteorological gauges at high elevations also increases the uncertainty in these regions. These findings
are consistent with previous efforts to quantify uncertainty in gridded precipitation datasets that found uncertainty between
150-200 mm/year in this region (Henn et al., 2018; Lundquist et al., 2015).

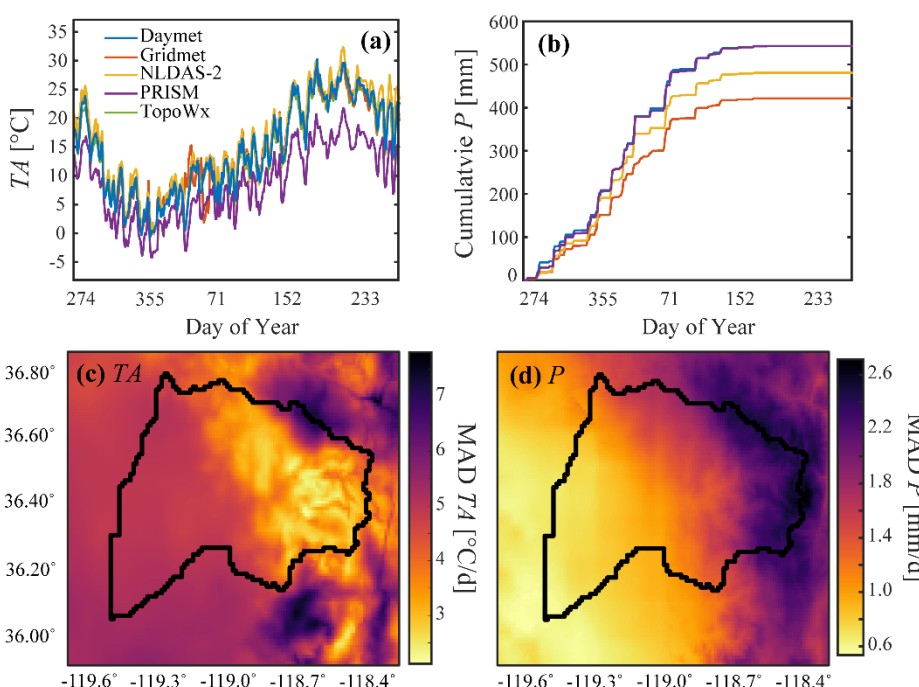

**Figure 3: (a) daily domain averaged values of air temperature for 5 temperature datasets, and (b) The cumulative sum**
**of domain averaged precipitation for each of the four gridded datasets in WY2016. Uncertainty in the daily air**





**temperature (c) and precipitation (d) datasets represented by the mean absolute difference (MAD) for the Water year 2016.**

## 3.2 Model Validation

A comprehensive validation of model performance is presented in Schreiner-McGraw and Ajami (2020). In this study
we present a validation of model performance in simulating two components of the energy balance, *ET* and *SWE*. Figure 4 presents a comparison between the simulated *ET* from each of the experiments 1, 2, and 3 and remote sensing values from the 8-day MODIS product. The values presented are watershed average values for the Kaweah River watershed with irrigated croplands removed due to the lack of irrigation in the simulations. Generally, the range of simulated monthly *ET* encompasses the remote sensing values. The peak value of monthly *ET* of 40 mm/mon is replicated by the model simulations. The timing
of the peak value, however, is inconsistent between the simulations and the remote sensing product. At the monthly scale, both the peak *ET* and the minimum *ET* throughout the year are delayed by 1 month. This result is partially attributable to the coarse temporal resolution of the remote sensing data composited at 8-day intervals, as well as the monthly aggregation of this data. In addition, we believe that some of the discrepancy arises from restricting the plant rooting depth in the simulations to the top 2 m of soil in ParFlow.CLM simulations, limiting their ability to draw on water stored in the saprolite layer. As saprolite
storage is recharged by spring snowmelt (Thayer et al., 2018), this model specification creates temporal discrepancy in *ET*. Because the simulated energy budget captures *ET* quantities, however, we are satisfied with the model performance considering the study objectives.





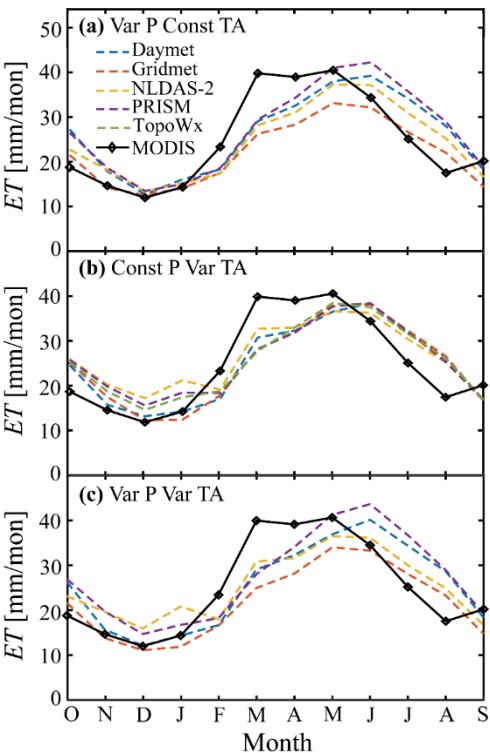

**Figure 4: Monthly *ET* in WY2016 from the MODIS remote sensing product (solid lines) as well as the range of simulated *ET* from each of the three experiments (a) VarPConstTA, (b) ConstPVarTA, and (c) VarPVarTA (dashed lines) in the Kaweah River watershed. Croplands are removed from this comparison as irrigation is not included in simulations.**

We also assess the performance of the energy budget simulations by comparing the simulated *SWE* to a reanalysis product developed for the Sierra Nevada region (Margulis et al., 2016). Figure 5 presents the annual cycle of snowpack accumulation and melting as simulated *SWE* from each of the three experiments. We present the total volume of *SWE* for each day in the Kaweah River watershed. For all the experiments, the simulated annual pattern of daily *SWE* encompasses the observed values. The only exception is the period from DOY 65-150 from the ConstPVarTA scenarios, where the simulated *SWE* is larger than the reanalysis product values (Fig. 5b). In the ConstPVarTA scenarios, significant variability within the simulated *SWE* exists, especially for the peak *SWE* values. The peak *SWE* of the Daymet scenario is 27% higher than the observed and *SWE* from the Gridmet scenario is 42% higher than the observed. The Daymet and Gridmet datasets have lower air temperatures in the mid-elevation zone where temperatures fluctuate between below and above freezing (Fig. 2). In terms of timing, the peak *SWE* occurs on DOY 74 for all ConstPVarTA scenarios except in the Daymet forcing scenario, where the peak *SWE* occurs on DOY 34. The timing of full snowmelt is more variable and is delayed for the scenarios with higher peak *SWE*. Full snowmelt occurs on DOY 216 for Daymet, DOY 221 for Gridmet, DOY 211 for NLDAS-2, and DOY 194 for the PRISM scenario. The simulated *SWE* from each of the VarPConstTA scenarios (Fig. 5a) has similar temporal patterns, but there is considerable spread in the *SWE* values that reflect the spread in precipitation volumes from the different forcing





datasets. The VarPVarTA scenarios have the largest variability in *SWE* across the forcing datasets, with NLDAS forcing underestimating the peak *SWE* and other forcings overestimating it relative to the observations (Fig. 5c).

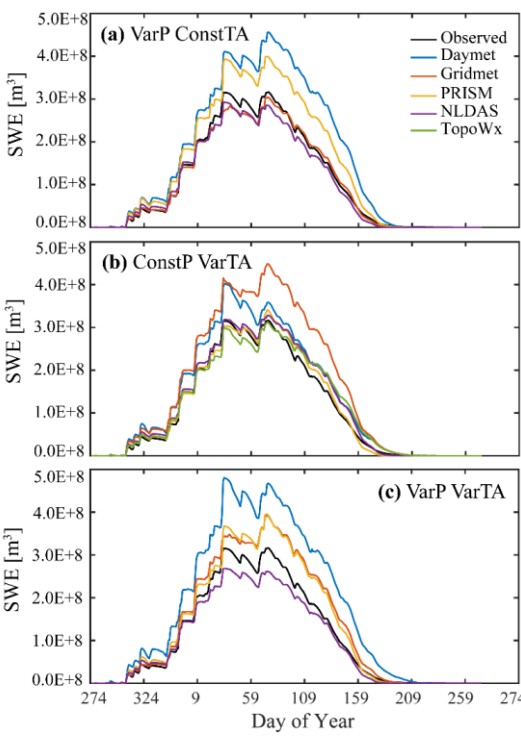

**Figure 5: Daily *SWE* from the reanalysis product (black lines) as well as the range of simulated *SWE* from each of the three experiments (a) VarPConstTA, (b) ConstPVarTA, and (c) VarPVarTA (color lines) in WY2016.**

### 3.3 Recharge Pathway Sensitivity to Meteorological Variability

Mountain system recharge to the Central Valley is a key unknown for water management in this highly productive agricultural system. Our simulations suggest that recharge rates from the Sierra Nevada mountain system to the Central Valley are significant. The total mountain system recharge from the mountainous portion of the Kaweah River watershed, the Kaweah Terminus sub-watershed, to the valley aquifer (*MBR* + *MFR*) ranges from 186–504 mm/yr, depending on which meteorological forcing scenario is used. In our simulations, the majority of this recharge comes from the *MFR* pathway, the ratio of *MFR*/(*MBR* + *MFR*) ranges from 0.85 to 0.99 across all simulations performed. Our results are consistent with observational studies (Visser et al., 2018), but there is considerable uncertainty related to characterizing the source of mountain system recharge. The simulated *MFR* depends on the subsurface permeability values assigned to the Central Valley aquifer in the piedmont slope region. Our hydraulic parameter values are based on drill core data and a previously calibrated hydrologic model (Faunt, 2009), but these values may be too high causing overestimation of simulated MFR (Brush et al., 2013). Historical observations under pre-development conditions suggest that the Kaweah River branched into several smaller distributaries, some of which did not flow all the way to the historic Kaweah Lake (U.S. EPA, 2007; Hall, 1886). These observations suggest that our *MFR* estimates





from the Kaweah River are reasonable, but are likely overestimated due to coarse horizontal model resolution resulting in streambeds that are unreasonably wide and potentially overestimated hydraulic conductivity of the Central Valley sediments. Conversely, the coarse resolution of the model may result in an underestimation of *MFR* via small channels and first-order watersheds located on the piedmont slope (Schreiner-McGraw and Vivoni, 2018).

Across all simulations, the total *MAR* (*MAR* from rain + *MAR* from snow) is dramatically larger than the *MBR*. This
is expected as the *MAR* is calculated as the potential recharge, and most of it may flow via local flow paths to topographically convergent zones where it could be subsequently transpired or discharged as baseflow; while the remainder becomes *MBR*. Figure 6 presents the range of simulated annual recharge from each of the mountain system recharge pathways. For all recharge pathways, the simulated value is impacted by the choice of temperature and precipitation datasets. The temperature datasets used in the ConstPVarTA scenarios result in a range of simulated recharge that is 16%, 24%, 3%, and 24% of the mean value
from the 5 scenarios for the *MAR* from rain, *MAR* from snow, *MBR*, and *MFR* pathways, respectively. The corresponding precipitation datasets included in the VarPConstTA scenarios result in a larger range in simulated recharge for all recharge pathways. The range of simulated recharge for the VarPConstTA scenarios is 26%, 52%, 240%, and 76% of the mean of 4 scenarios for the *MAR* from rain, *MAR* from snow, *MBR*, and *MFR* pathways, respectively. When variability in *TA* is added to *P* variability in the VarPVarTA scenarios, the range of simulated recharge for each pathway increases to 33%, 70%, 238%,
and 91% of the mean of the four VarPVarTA scenarios, for the *MAR* from rain, *MAR* from snow, *MBR*, and *MFR* pathways, respectively.

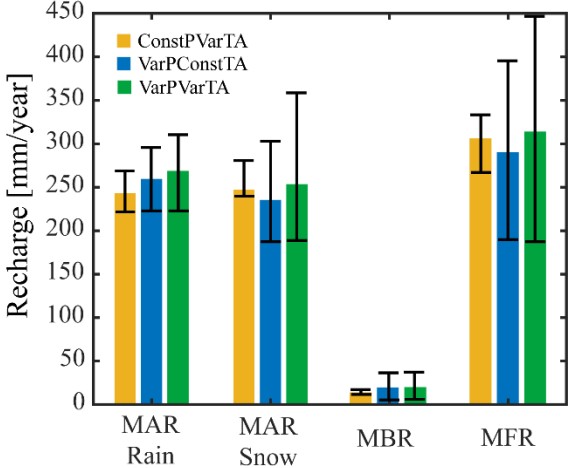

**Figure 6: Mean and standard deviation of simulated mean *MAR* from rain and snow, mountain block recharge (*MBR*), and mountain front recharge (*MFR*) from scenarios in three simulation experiments: ConstPVarTA, VarPConstTA**
**and VarPVarTA in the Kaweah Terminus watershed.**

To compare the sensitivity of each mountain recharge pathway to changes in meteorological forcings, we calculate the ε and *S* for different recharge pathways in the Kaweah Terminus watershed. Figure 7 displays the average ε and *S* across the four forcing datasets, for each of the four mountain recharge pathways. For changes in *P* forcing, *MBR* and *MFR* are more





sensitive to the forcing dataset than both components of *MAR*, while the rainfall-*MAR* is the most sensitive to changes in *TA*
dataset. The rainfall-*MAR* is least sensitive to changes in *P*, while the snow-*MAR* is least sensitive to changes in *TA*. This result
in part is a reflection of the higher mean *MAR* values that makes changes relative to the mean value smaller. Additionally,
most of the precipitation uncertainty is in the high elevation zone where temperatures are low across all forcing datasets, and
snow is dominant. As a result, although we might expect snow-*MAR* to be highly sensitive to changes in *TA*, it is much more
sensitive to changes in *P*. Following the same logic, each of the three recharge pathways that is controlled by *SWE* (Snow-
derived *MAR*, *MBR*, and *MFR*) is more sensitive to changes in *P* than changes in *TA* (Fig. 7). Rainfall-*MAR* has the highest
sensitivity to *TA* uncertainty compared to snow-*MAR*, *MFR* and *MBR* (Fig. 7a) as temperature controls partitioning of
precipitation to snow and rainfall. For *P* uncertainty, the ε of *MFR* and *MBR* are higher than the ε of the two *MAR* components
(Fig. 7b), although ε of the *MBR* is influenced by the smaller magnitude of *MBR*. This result is consistent with our
understanding of the soil water budget, *MFR* requires the generation of overland flow, and *MBR* requires deep percolation of
soil water through fractured bedrock. Both of these processes require precipitation excess and should be influenced by
precipitation variability.

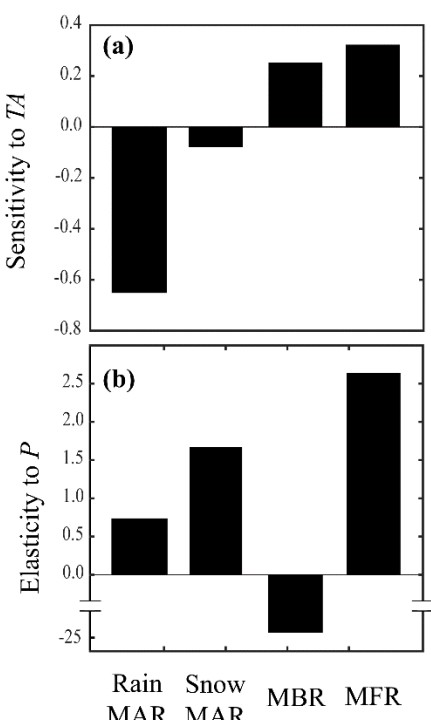

**Figure 7: Sensitivity to TA (a) and elasticity to P (b) of different mountain system recharge pathways. Each bar**
**represents the mean value for the scenarios in the ConstPVarTA (a) and VarPConstTA (b) experiments in the**
**Kaweah Terminus watershed.**

Prior efforts have demonstrated that topography driven subsurface flow is an important process that redistributes
uncertainty in *P* forcing throughout the watershed (Schreiner-McGraw and Ajami, 2020). Figure 8 presents the relations





between *TWI* and the uncertainty in simulated *MAR* (σ*MAR* - defined as the standard deviation of recharge across the scenarios in each experiment) for the Kaweah Terminus watershed. We limit this analysis to the Kaweah Terminus watershed because

it has the same vegetation type (evergreen forest) and relatively consistent climate conditions to make the *TWI* a valid expression of the topographic effect on soil water movement. By limiting the analysis to the mountainous region, the potential recharge is equivalent to our definition of *MAR*. Across all experiments, the uncertainty in *MAR* increases with *TWI* because topography driven flow moves water into convergent zones via lateral soil and shallow groundwater fluxes. An ANCOVA test reveals that the strength of the topographic control on *MAR* uncertainty is higher for the ConstPVarTA scenarios than the

VarPConstTA scenarios, as represented by the statistically significant higher slope (Fig. 8a,b). This result, along with higher soil moisture values (data not shown), suggests that the *ET* in convergent zones is more energy limited than water limited throughout the year, so *TA* uncertainty creates larger variability in *ET* than *P* uncertainty. Due to the link between *ET* and potential recharge via soil wetness, this variability in *ET* is reflected in increases in *MAR* variability. When uncertainty in both *TA* and *P* is considered, the slope of the *TWI* and σ*MAR* relation increases, but according to an ANCOVA test, the slope is not

significantly different ($p < 0.05$) than the ConstPVarTA scenarios. Because topography driven subsurface flow concentrates soil water in convergent zones, the individual spatial patterns of *P* and *TA* uncertainty become less important and their uncertainties cancel each other out, creating consistently negative interaction effects in the VarPVarTA scenarios. This impact is more pronounced with *MAR* compared to other variables because *MAR* is the most dependent variable on topography driven flow.





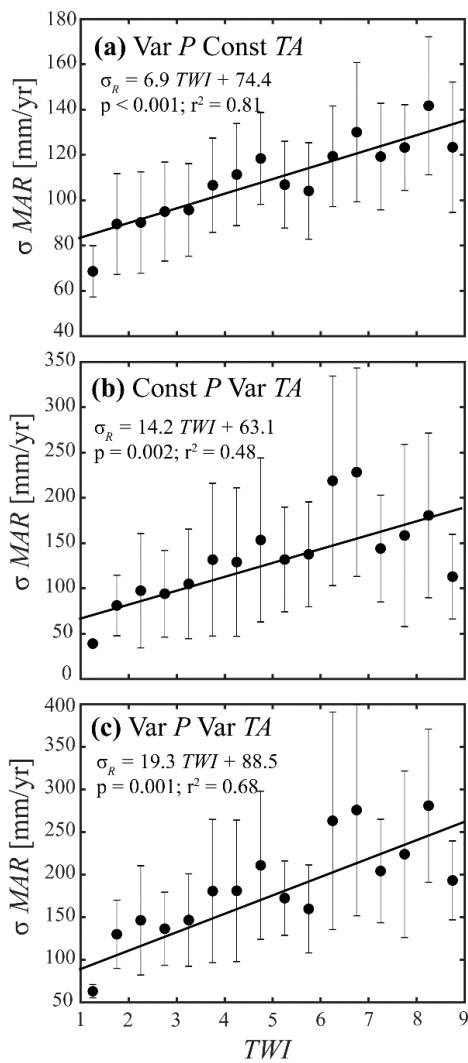

**Figure 8: Scatterplots between the binned values of *TWI* and the standard deviation of *MAR* from each of the scenarios included in the VarPConstTA experiment (a), the ConstPVarTA experiment (b), and the VarPVarTA experiment (c) in the Kaweah Terminus watershed. Circles represent the bin average and the bars represent the bin's standard deviation. Solid lines present statistically significant (p<0.05) linear regressions.**

### 3.4 Uncertainty of Water Budget Partitioning to Meteorological Forcings

The *P* and *TA* simulation scenarios in experiments 1 and 2, respectively, allow assessing the relative importance of uncertainty in *P* and *TA* on the simulated water budget. Figure 9a presents the simulated annual water budget partitioning for the mountainous Kaweah Terminus watershed for all of the scenarios in experiments 1, 2, and 3. The Kaweah Terminus watershed is used because it is the largest sub-watershed in the domain where accurate streamflow simulations can be ensured





through model validation. Variable precipitation forcing applied in experiment 1 (VarPConstTA) results in significant changes to the water budget partitioning. For all $P$ forcing datasets (ConstTAVarP scenarios), $MBR$ remains the smallest portion of the water budget, while $ET$ composes the largest portion of the water budget. The largest changes in the water budget partitioning occur in the simulated $Q$ that ranges from 28% to 46% of the precipitation. Changes to the $TA$ forcing dataset when the

precipitation is constant (ConstPVarTA scenarios), result in similar patterns as changes to the $P$ forcing when the temperature is constant. For all $TA$ datasets, $ET$ is the largest component of the water budget and $MBR$ is the smallest. The variable $TA$ scenarios result in a smaller range of simulated $Q$ (36-45%) than the variable $P$ datasets (28-46%), but a larger range in simulated $ET$ (46-54% for VarPConstTA and 44-53% for ConstPVarTA). The right-most column in Figure 9a presents the water budget partitioning when both $TA$ and $P$ forcing datasets are varied. There is considerable uncertainty in the major water

budget components, and when both Daymet $P$ and $TA$ are used, the water budget shifts so that $ET$ is no longer the largest component. The $ET$ ranges from 39% (Daymet) to 56% (downscaled NLDAS-2) while the $Q$ ranges from 25% (Gridmet) to 44% (Daymet) of the total water budget. These ranges are much larger than the range in water budget partitioning caused by modifying $P$ or $TA$ individually, and suggests that the uncertainty from the individual forcing variables is additive, rather than cancelling each other out. Besides $P$ and $TA$ uncertainties, differences in the water budget partitioning of VarPVarTA scenarios

are due to non-linear feedbacks between the spatial patterns of $P$ and $TA$ and subsurface properties, vegetation type, and topography.

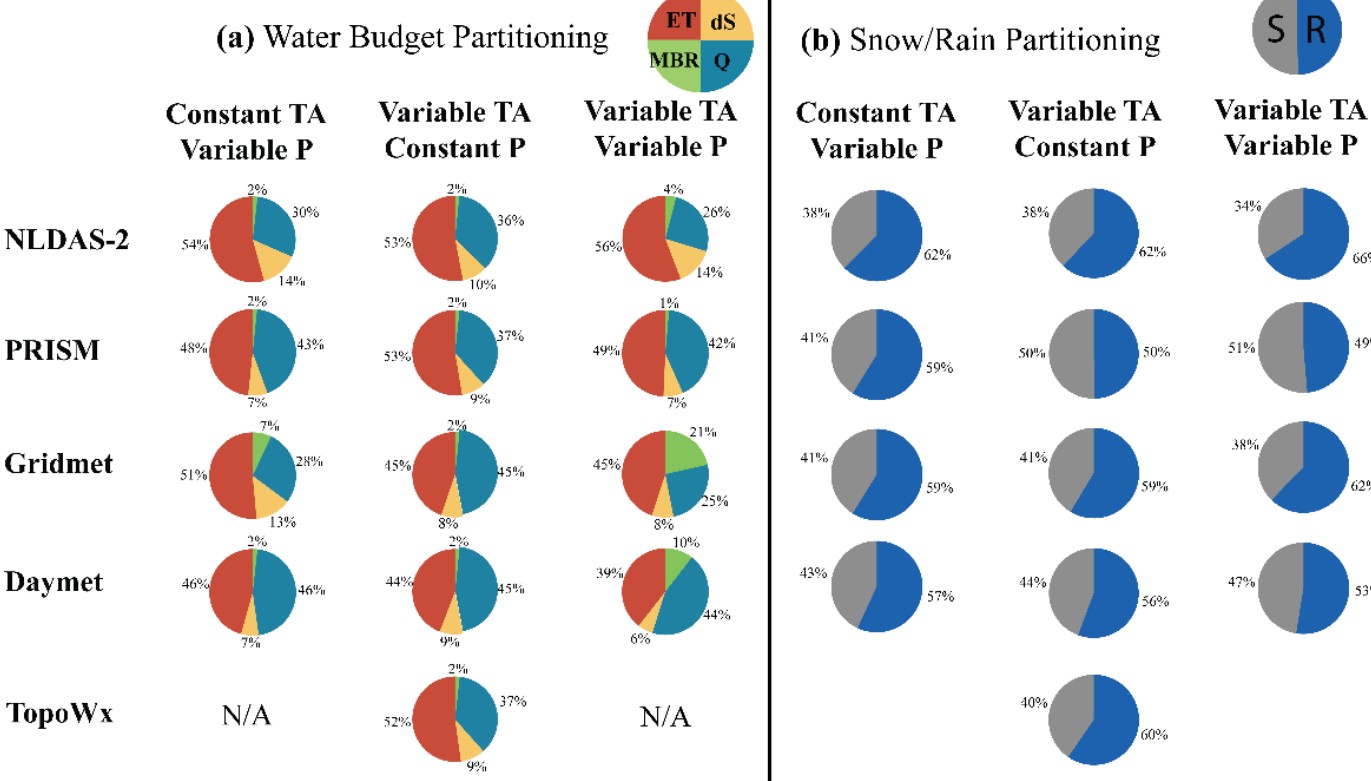





**Figure 9: (a) Water budget partitioning shown as a fraction of the incoming annual precipitation for the Kaweah Terminus watershed, and (b) rain/snow partitioning in the Kaweah Terminus watershed for each of the scenarios in the three simulation experiments. Fractions are rounded to the nearest 1%.**

Figure 9b presents the proportion of the total precipitation that falls as snow or rain for each scenario from experiments 1, 2, and 3. Changes to the *TA* forcing dataset create a larger range in the snowfall/*P* ratio than changes to the *P* forcing dataset (snowfall/*P* ratio of 38-43% in VarPConstTA and 38-50% in VarTAConstP). A close inspection of the charts presented in Figure 9 suggests that the snow/rain ratio impacts the annual water budget partitioning, and Figure 10a-c demonstrates this conclusion by presenting relations between the ratio of snowfall/*P* and the *ET*/*P*, *MBR*/*P*, and *Q*/*P* ratios. Each point in Figure 10 represents the mean value for each scenario in experiments 1, 2, and 3. Statistically significant linear relations (p<0.05) demonstrate that an increase in the proportion of *P* that falls as snow, decreases the *ET*/*P* ratio and increases the *Q*/*P* ratio. As the ConstPVarTA scenarios create a larger range in snowfall/*P* ratio than the VarPConstTA scenarios (Figure 9b), this raises the question of why the ConstPVarTA scenarios do not create a larger range in simulated *MBR* or *Q*? Although there are significant relations (p < 0.05) between the Snowfall/*P* ratio and water budget partitioning, the relations are weak with $r^2$ values between 0.17 and 0.35. Figure 10d-f presents the relations between the total annual snowmelt ($S_m$) and the *ET*/*P*, *MBR*/*P*, and *Q*/*P* ratios. The $S_m$ has stronger relations with the water budget partitioning than the snowfall/*P* ratio with $r^2$ values of 0.68-0.79. In the mountainous study watershed, the total volume of snowmelt is more dependent on *P* than *TA* because the high elevation regions where the majority of the precipitation falls remain below freezing for most of the wet season across all air temperature datasets. The increased variability in total snowmelt results in the larger changes to *Q* and *MBR* caused by the VarPConstTA scenarios.

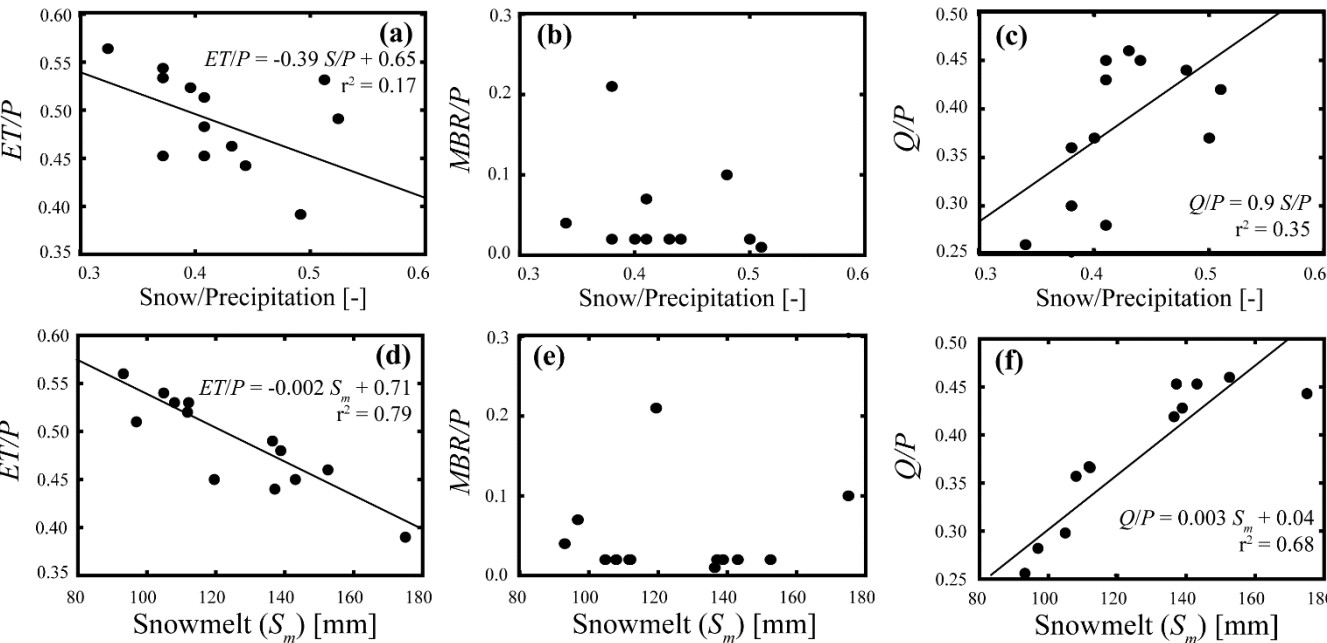





**Figure 10: Scatterplots illustrating the relation between the annual snow/precipitation ratio and *ET/P* (a), *MBR/P* (b), and *Q/P* (c) ratios. We also present the relation between the total annual snowmelt (*S~m~*) and the *ET/P* (d), *MBR/P* (e),**

**and *Q/P* (f) ratios. Each point represents the value for the Kaweah Terminus watershed from each of the forcing scenarios in experiments 1-3. Black lines represent statistically significant (p<0.05) linear relationships.**

In the Mediterranean climate of California, the distinct dry season creates challenges for water management, making the temporal patterns in simulated water budget variability of interest. Figure 11 presents the monthly time series of *ET*, *MAR*, and *Q* from the Terminus watershed for the base case scenario (solid lines) with the range of simulated values (dashed lines),

as well as the relative range for each (black bars). The variable *P* forcings, from the VarPConstTA scenarios, result in a relatively consistent monthly *ET*-based $R_r$ throughout the year. On average, the *ET*-based $R_r$ is 0.2 throughout the year and January (0.3) and February (0.1) are the months with the largest discrepancies. Changes in the *P* forcing dataset cause larger variability in the $R_r$ for the *Q* and *MAR*, but a seasonal pattern does not emerge. Scenarios with altered *TA*, however, display a more prominent annual trend in the $R_r$ of simulated *ET* and *Q*. The *ET*-based $R_r$ is considerably higher in November, December,

and January for the ConstPVarTA scenarios (average $R_r$ is 0.5), compared to 0.07 for the rest of the year. This finding is striking because the divergence in the *TA* forcing datasets is primarily found during the summer months (DOY ~150 – 230) (Fig. 3c). We attribute this result to the fact that *ET* does not occur if the temperatures are below freezing, and *TA* variability at a given location may result in below freezing temperature for one *TA* dataset, but not another. The *Q*-based $R_r$ increases during March through July consistent with the snowmelt period and increases in *TA* variability (Fig. 3). For VarPVarTA

experiments the *MAR*-based $R_r$ varies throughout the year with higher values in the months of July and October, although the trend is not dramatic.

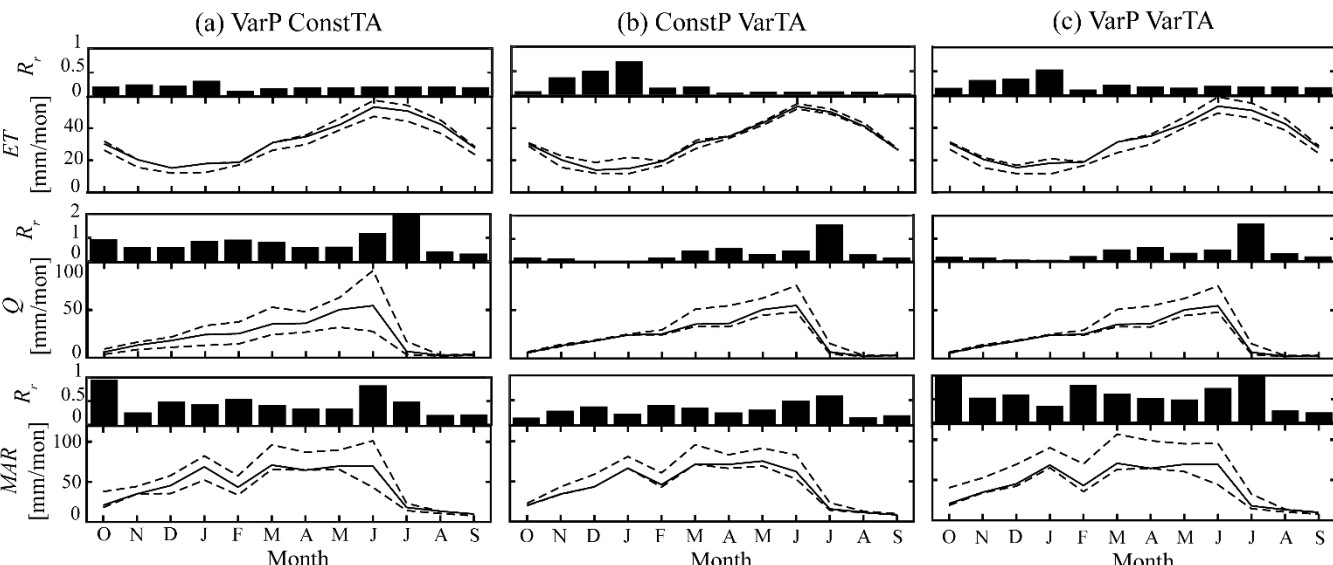

**Figure 11: Monthly values of *ET*, *Q*, and *MAR* from the Kaweah Terminus watershed are presented for each of the three experiments (a) VarPConstTA, (b) ConstPVarTA, and (c) VarPVarTA. The solid lines represent the values from**





**the base case scenario while the dashed lines present the range of values from the scenarios included in each experiment. Bars represent the relative range ($R_r$), defined as the range of simulated values for each experiment divided by the monthly value from the base case.**

### 3.5 Sensitivity and Elasticity of Simulated Water Budget to Precipitation or Air Temperature

     In addition to a close examination of the water budget, we calculate the elasticity ($\varepsilon$) and sensitivity ($S$) of water

budget components to changes in $P$ or $TA$, respectively, over the entire Kaweah River watershed. Figure 12 presents the $\varepsilon$ and $S$ calculated for each meteorological forcing scenario in experiments 1 and 2, relative to the base case, for the water budget components. In general, results suggest that the water budget is very sensitive to changes in forcing. Elasticities are larger than 1 for most datasets and variables such as $SWE$, $dS$, potential recharge ($R$), and $Q$ at the Terminus dam indicate that the simulated variables exhibit a larger percent change than the percent change in precipitation. The sensitivity of the simulated water budget

to changes in temperature is also quite high, especially when the Gridmet and downscaled NLDAS-2 datasets are used (Fig. 12b). The only hydrologic variables that are not heavily impacted by changes in $P$ or $TA$ are the land surface temperature ($T_g$) and root zone volumetric water content ($\Theta$). At the annual scale, this result is not surprising because the soil moisture is depleted by $ET$ and $R$ and its variability is highest on a daily rather than annual scale.

     For the majority of variables examined, the sensitivity to changes in $TA$ is relatively high partially because the

watershed average changes to $TA$, the denominator in Eqn. (2), are typically low, while the spatial patterns of $TA$ exhibit larger differences. Sensitivity of the Gridmet and downscaled NLDAS-2 datasets to changes in $TA$ have opposing signs for all simulated variables. This result is likely because the spatial patterns of $TA$ have the opposite differences from the base case (Fig. 2); Mean annual temperature from Gridmet (NLDAS-2) is warmer (cooler) in the mountains and cooler (warmer) in the low elevations. Overall, the water budget exhibits high $\varepsilon$ and $S$ to both changes in $P$ and $TA$. This behaviour does not necessarily

mean that the magnitude of $P$ and $TA$ effects on the water budget are equal. It means that the range of uncertainty contained in the meteorological forcing datasets for both $P$ and $TA$ results in similar amounts of uncertainty in the simulated water budget.





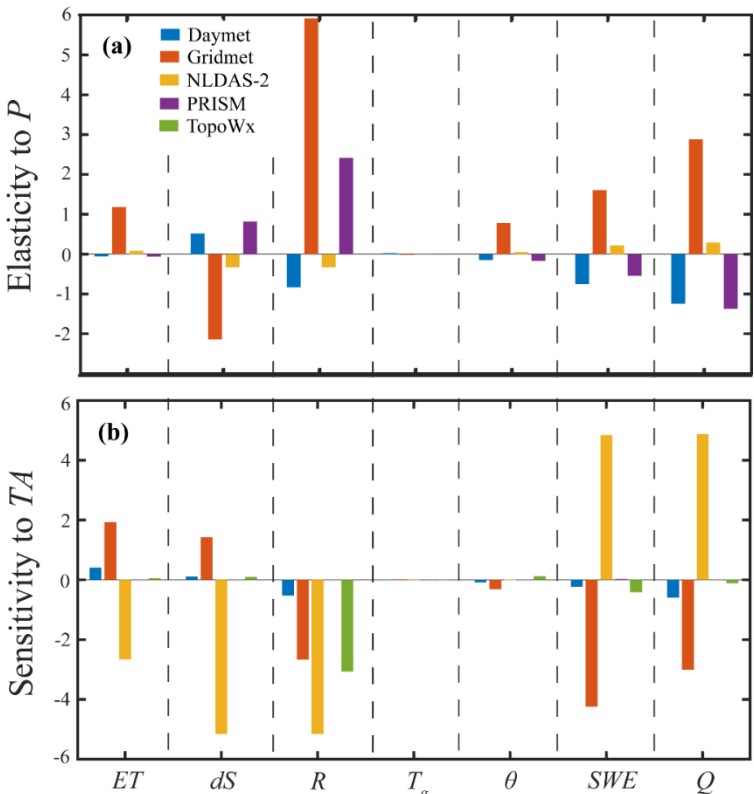

**Figure 12: Elasticity (a) and sensitivity (a) of simulated hydrologic variables evapotranspiration (*ET*), change in subsurface storage (*dS*), potential recharge (*R*), land surface temperature (*T_g*), root zone soil moisture (*Θ*), snow water equivalent (*SWE*), and streamflow (*Q*), to variability in precipitation and air temperature. Each bar represents the average value from the Kaweah River watershed, except streamflow measured at the Kaweah Terminus dam. Elasticities were calculated using the scenarios from the VarPConstTA experiment and sensitivities were calculated using the scenarios from the ConstPVarTA experiment.**

Besides watershed average sensitivities, we also calculate the sensitivity and elasticity of variables for each pixel in the Kaweah River watershed. As an example, we present the ranges in $S$ and $\varepsilon$ for potential recharge (*R*) at the pixel scale in Figure 13. Despite differences in the spatial patterns of *TA* across the domain for different datasets (Fig. 2), the ranges in annual *R* sensitivities are very small particularly for the PRISM dataset. The PRISM *TA* dataset has a zero or cold bias relative to the mean dataset while the bias in other datasets ranges between -8 to 8 C. Ranges of *R* elasticities across the Kaweah River watershed are not uniform and Gridmet and downscaled NLDAS-2 datasets have the largest ranges in *R* elasticities. These datasets have the lowest mean annual precipitation and the ratios of *ET*/*P* are the highest compared to other datasets (Fig. 9a). While variability in *R* elasticities are large, any spatial patterns are difficult to discern from spatial maps and elasticities are not controlled by topography, forcings, subsurface properties, or vegetation type.





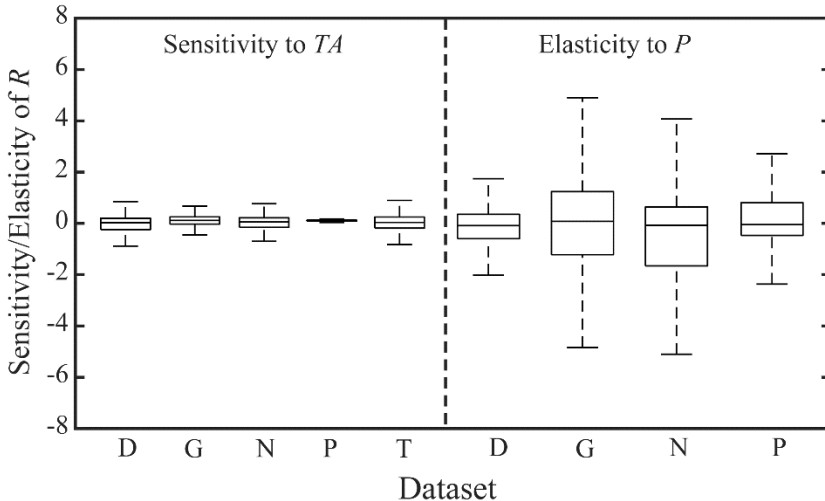

**Figure 13: Box-plot of sensitivity (*S*) and elasticity (*ε*) of potential recharge (*R*) calculated at the pixel scale for the**
**Kaweah River watershed. The sensitivity to *TA* is calculated using the scenarios from the ConstPVarTA simulations**
**and the elasticity to *P* is calculated using the scenarios from the VarPConstTA simulations. D=Daymet; G=Gridmet;**
**N=downscaled NLDAS-2; P=PRISM; T=TopoWx.**

**3.6 Interaction Effects of Combined Changes to Precipitation and Air Temperature on the Water Budget**

Understanding the individual impacts of uncertainty in *TA* and *P* forcings provides a foundation for how to manage
uncertainty in meteorological forcings. But, as climate change is expected to alter both air temperature and precipitation, it is
important to understand how uncertainty in both datasets combines to alter the simulated water budget. To test the extent to
which the two sources of uncertainty superimpose, we compare the differences between hydrologic variables simulated with
the base case scenario to simulations that alter both *TA* and *P* (VarPVarTA, experiment 3). We use the Kaweah River watershed
for these calculations. This comparison is done for the average value of each hydrologic variable over Kaweah watershed,
while *Q* is represented at the Kaweah Terminus dam. Figure 14a displays the estimated changes to the simulated hydrologic
variables ($v_{\Delta P\Delta TAest}$), relative to the base case scenario, if the impact of uncertainty from the ConstPVarTA and VarPConstTA
scenarios were additive. Fig. 14b displays the actual changes caused by the VarPVarTA simulations, and Fig. 14c presents the
difference between the estimated and actual changes. The difference can be interpreted as the strength of the interaction effects,
i.e. a difference of 0.05 indicates that the interaction effects between *TA* and *P* increased the value of the variable, *v*, by 5%.
Generally, the differences between estimated and simulated values are quite small, suggesting that the interaction effects
between *TA* and *P* uncertainty are small. Indeed, the majority of interaction effects are between -5% and 5%. The primary
exception to this pattern is found in the variables related to groundwater, *dS* and *R*. The *dS* is the simulated variable with the
largest variability in the interaction effects. For example, the Gridmet dataset results in interaction effects of -40% while the
PRISM dataset results in interaction effects of 3% in changes in subsurface storage. With the exception of the PRISM dataset,





the interaction effects for *dS* are all negative. Additionally, across all four datasets the interaction effects decrease *R*, with an
average value of -5.1%. This is because an increase in *P* will generally increase *R* while an increase in *TA* will generally
decrease *R* by increasing *ET*. This effect can be exacerbated by topography driven flow that concentrates soil moisture in
convergent zones.

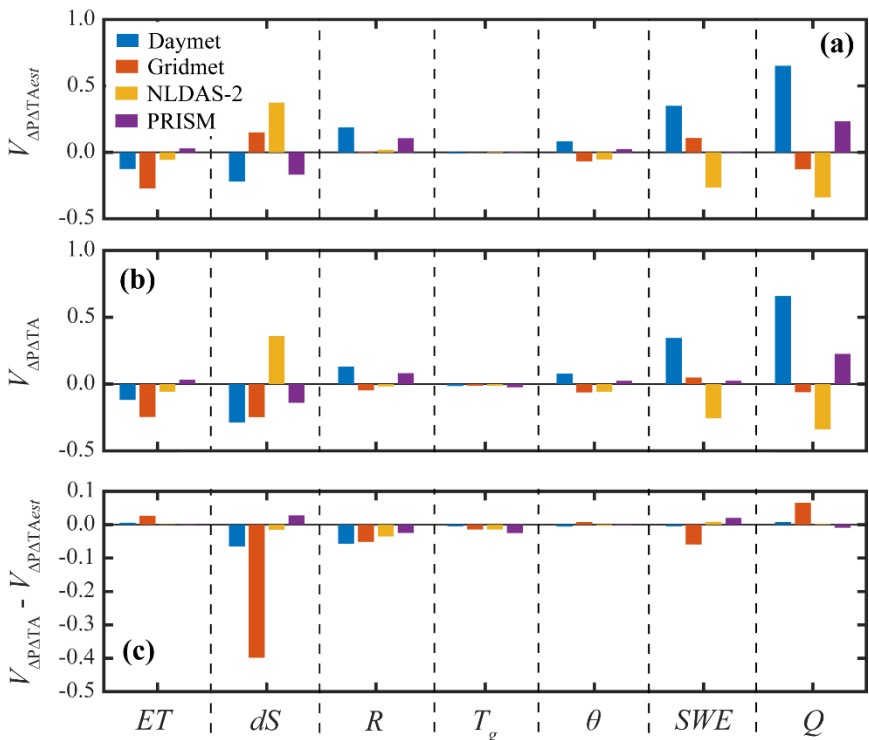

**Figure 14: Impacts of combined TA and P on simulated hydrologic variables in the Kaweah River watershed. (a) The
estimated relative difference in the hydrologic variables evapotranspiration (*ET*), change in subsurface storage (*dS*),
potential recharge (*R*), land surface temperature (*T_g*), root zone soil moisture (*Θ*), snow water equivalent (*SWE*), and
streamflow (*Q*), if the effects of air temperature and precipitation changes are linearly additive. (b) The relative
difference between the base case and each of the VarPVarTA scenarios. (c) The difference between the predicted and
actual changes from combined variability in *P* and *TA*.**

## 4 Summary

In this paper, we examine the propagation of uncertainty in the meteorological forcings, precipitation and air
temperature, into groundwater recharge simulated with the integrated hydrologic model, ParFlow.CLM. We use the Kaweah
River watershed as a study domain to (1) quantify groundwater recharge from the mountain system, and assess which recharge
pathway is most sensitive to meteorological variability, (2) determine whether uncertainty contained in common *P* or *TA*
gridded datasets has a larger impact on the simulated water budget, and (3) evaluate the strength of interaction effects when




both *P* and *TA* are uncertain. In the course of this analysis, we perform three sets of model experiments by altering forcing datasets to compare to our base case scenario forced with the mean *P* and mean *TA*. These experiments include variable *P* constant *TA* (VarPConstTA), constant *P* variable *TA* (ConstPVarTA), and variable *P* variable *TA* (VarPVarTA).

Given that the *P* datasets differ in their total annual precipitation by 30% (125 mm), and variability in the spatial distribution of precipitation is large, one might expect that the choice of *P* dataset would be more important than the choice of *TA* dataset. Our analysis revealed that in a mountainous system, the impact of uncertainty in gridded *P* datasets is similar to the impact of uncertainty in available *TA* datasets. The range of values in the simulated water budget partitioning for the VarPConstTA scenarios and the ConstPVarTA scenarios are comparable. This result is attributed to the impact of air

temperature on snow processes. Variability in *TA* creates variability in the partitioning of precipitation into rain and snow. This partitioning alone impacts the water budget where higher ratios of snow/rain results in more potential recharge. Additionally, air temperature impacts the snowmelt rate and the total amount of snowmelt is a strong control of the water budget partitioning, with higher snowmelt leading to less *ET* and more potential recharge, which is discharged from the mountain system into streamflow. We calculate the sensitivity and elasticity of changes in the water budget to changes in *TA* and *P*, respectively.

We find that groundwater recharge and storage changes are highly sensitive to both changes in *TA* and *P*. Our results demonstrate that the high levels of uncertainty in both *TA* and *P* gridded datasets have profound impacts on the water budget simulated by an integrated hydrologic model where surface and subsurface processes are coupled.

        The uncertainty in the simulated water budget caused by the separate uncertainty in *TA* and *P* forcing datasets is largely superimposed when the model is forced with variable *TA* and variable *P*. For most water budget components, the

interaction effects of *TA* and *P* uncertainty reduce the combined impact of uncertainty by less than 5%, i.e. the variability in the simulated water budget caused by combined changes to *TA* and *P* forcing is within 5% of the sum of the variability from individual changes. The exception to this result is found in the groundwater system. Potential groundwater recharge and changes in subsurface storage exhibit larger interaction effects than the surface water budget. This is attributed to the role of topography in controlling lateral subsurface flow in the shallow groundwater system. The uncertainty in groundwater recharge

rates is highest in regions of convergent topography for all three experiments. But the uncertainty in these regions is much higher when variable *TA* forcings are used. This is because the topography concentrates water in these locations so that *ET* becomes energy limited. As a result, variability in *TA* creates more variable *ET* and recharge.

        Finally, all of the recharge pathways present in the mountainous Kaweah watershed, *MAR*, *MBR*, and *MFR*, are more sensitive to changes in *P* than changes in *TA*. It should be noted, however, that comparisons are difficult due to different units

for *P* and *TA* sensitivities. The higher sensitivity to *P* dataset is because these pathways largely depend on snowmelt, and precipitation is concentrated in the winter at high elevation regions where the air temperature remains well below freezing during this time period. The *MAR* pathway is less sensitive to changes in *P* than the other pathways, particularly when *MAR* is derived from rainfall. Our simulations suggest that mountain system recharge to the Central Valley aquifer is a significant portion of the water budget regardless of the meteorological forcing dataset used. Indeed, *MFR* contributes between 186 and

504 mm/yr of recharge from the Kaweah Terminus watershed to the Central Valley aquifer. A large fraction of the Kaweah



Terminus watershed water budget (25-46%) becomes *MFR* in the Central Valley region. In our simulations, *MFR* is the primary pathway via which the mountain system recharges the Central Valley aquifer, accounting for 85-99% of the total recharge. The high uncertainty in subsurface geologic structure and parameters, however, creates large uncertainties in the quantities of *MBR*. Overall, the results from this study highlight the importance of uncertainty in forcing datasets when simulating the
groundwater response to climate change. The magnitude of simulated changes in the groundwater recharge due to meteorological forcing uncertainty highlights the need for hydrologists to improve gridded datasets to improve our understanding of how meteorological variability propagates into groundwater in topographically complex mountain systems.

## Acknowledgements

This research is funded by the California Energy Commission grant "Advanced Statistical-Dynamical Downscaling Methods and Products for California Electrical System" (award no. EPC-16-063), and the National Science Foundation CAREER Award (award no. 1944161). All datasets used and model code are publically available. Model outputs will be deposited at UC Riverside Dryad Digital Repository, https://datadryad.org/stash.

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
