# Peer review of "Combined Impacts of Uncertainty in Precipitation and Air Temperature on Simulated Mountain System Recharge from an Integrated Hydrologic Model"

_Hydrology and Earth System Sciences, 2020_

## Author Comment (AC1)

**Response to Reviewer #1:**

First of all, we would like to thank reviewer #1 for his/her comments on the paper. Their effort has helped us to improve the manuscript and we appreciate you agreeing to review the paper during these challenging times. Here, we provide point-by-point responses to each of reviewer 1's comments.

| Reviewer 1 comments | Author response |
|---|---|
| It needs to be demonstrated how/if the results of this study depend on the single year of the simulation. Why not extend the study period to include the last 30 years or so? All of the forcings used in this study go back to early 1980s. Even MODIS goes back to the early 2000s. A longer period of analysis can also help address the question of impact of forcing uncertainties on the long-term changes in mountain system recharge, which would be of interest given the focus on global warming driven changes in precipitation and temperature. A longer analysis period could also allow for independent verification of the mountain system recharge simulations such as by using GRACE based estimates of recharge, which goes back to early 2000s. This could help identify the set of atmospheric forcings which yield the most realistic estimates of the recharge. | You are correct, thank you. The model requires hourly forcing data and the spatially downscaled, hourly forcing that we use (Princeton CONUS Forcing) is only available from 2002. During this period, California suffered from the worst drought in recorded history. Our initial thoughts were that if we used this period, it would likely bias the model results based on these extreme years. But, as you state, it would make the findings more robust.

To address reviewer comment in the revised manuscript, we will add two more years of simulation in addition to the water year (WY) 2016 originally selected. Please note that (WY) 2016 approximately represents average precipitation and air temperature in the watershed. To assess the impact of hydroclimatic condition on our results, we plan to perform simulations for WY2014 and WY2011, representing extreme dry and wet years, respectively. Similar to our approach for WY2016, we will evaluate model simulations results at equilibrium to remove the impact of initial condition bias on results. Integrated hydrologic models like ParFlow.CLM model are particularly sensitive to initial conditions, which is why we chose to analyze simulations that had reached equilibrium conditions for each water year. It is not feasible to allow a 15-30 year simulation reach equilibrium because of the computational demands of the model. Therefore, we plan to perform simulations for the select dry and wet water years at equilibrium conditions to assess generality of |

| | our results. While using GRACE data is ideal for confirming changes in terrestrial water storages, the resolution of GRACE data is too coarse for the study basin. Furthermore, we do not include irrigation and water management options in this version of the model so it is not possible to assess the impacts of forcing uncertainty over the entire basin. |
|---|---|
| Additionally, how are the results of this study dependent on the choice of the hydrologic model? As shown by Vano et al, 2012 (cited by this manuscript too) depending on the choice of hydrologic model sensitivity of hydrologic variables (such as runoff) to changes in precipitation and temperature can vary substantially. | We agree that the choice of hydrologic model will impact our results. In the revised manuscript we will make it clear that our results are applicable to integrated surface water – groundwater models that implement 3D Richard's equation to simulate variably saturated subsurface flow across the entire subsurface, and have a fully integrated overland flow simulator. Of course, different model physics will result in different sensitivities. However, we are using the most physically-based approach for simulating surface water-groundwater processes, and have done detailed model validation to make sure major hydrologic processes are captured by the model. Ideally, one should perform such simulations using different model structures to assess the impact of all uncertainty sources on simulated hydrologic response.

We will add text to discuss the role of model selection in our results. |
| Finally, it also should be at least discussed how the results of this study may depend on the choice of the study domain. | Yes, this is a good point. Model parameterization and geologic setting are likely play a major role in how uncertainty in meteorological forcings will propagate into groundwater. In the revised manuscript, we will expand upon this discussion. For example, our simulations are performed in a mountain region underlain by fractured, low permeability bedrock. Previous work has shown that groundwater in these regions responds quickly to changes in precipitation (Pfister et al., 2017), which would likely impact the results. In the revised manuscript we will include discussion of how the |

| | geologic setting of our study site, and parameterization choices that we made are likely to impact the study results. While the role of uncertainty in precipitation forcing is discussed extensively, our main goal here was to highlight the role of temperature in addition to precipitation for regions with high relief. Of course, the obtained sensitivities in different mountain settings are impacted by the quality of meteorological forcings, topography, vegetation and subsurface characteristics. |
|---|---|
| I am surprised a bit about the differences in the simulated variables generated using GridMET, NLDAS and PRISM datasets. As described in Abatzoglou 2013, GridMET is based on the NLDAS-2 and PRISM dataset. Please at least discuss why this might be the case. | We agree that it is worth further highlighting these differences in the paper. We believe that the differences among products are caused by the fact that we are using different versions of PRISM and NLDAS-2 than the version used in Abatzoglou (2013) paper to generate the Gridmet dataset. To build the Gridmet dataset, they used the 800 m resolution version of PRISM, while we used the freely available 4 km resolution of PRISM data. Additionally, we used a downscaled version of the NLDAS-2 dataset, called the Princeton CONUS Forcing dataset with ~3 km resolution. As described in the paper, the Princeton dataset is the downscaled version of the original NLDAS-2 data with ~12 km resolution and the rainfall data is updated by using the radar products. We believe that the differences in the resolution of the dataset and interpolation approach have caused the differences in precipitation forcing datasets.

In the revised manuscript we will add this information to clarify. |
| Page 1, line 28: "high qualify" should be "high quality" | Thank you for pointing this out, it will be fixed in the revised manuscript. |
| Page 8, 217-219, how does this chosen threshold of 2.5 deg C for partition of precipitation into rainfall and snow, affect the results of this analysis, especially in the mid to low elevation parts of the domain? | To be clear, this threshold was not chosen by us, it is the threshold that the CLM model uses to partition precipitation into rainfall and snow. That being said, this threshold likely impacts the results. However, we did not assess its impacts. |

| | In the mid to low elevation portions of the domain, where precipitation can currently fall as either rain or snow, the snow melts quickly and snowpack does not accumulate due to higher temperatures in mid-elevation regions. We will add discussion to the revised manuscript, in the same location where we discuss the effect of model formulation on the results, to include the impact of temperature threshold to partition between rain and snow. |
|---|---|
| Page 22, lines 496-498, I am not sure why land surface temperature and soil moisture would not be affected by the choice of forcings, wouldn't changes in ET affect both? Please clarify. | Yes, changes in ET does affect both land surface temperature and soil moisture. At the annual scale, however, changes in soil moisture are small because changes in ET can be balanced out by changes in potential recharge and lateral soil moisture redistribution. In the revised manuscript, we will clarify the text in this section.

We believe that lower sensitivity to land surface temperature is partly related to the simplification made to represent the ground heat flux calculation in CLM. Many land surface models, including CLM, only incorporate heat transport via conduction and this simplification decouples heat transport from soil moisture transport. Including heat convective transport through soil moisture distribution will increase computational time. While the ParFlowE model (Kollet et al., 2009) incorporates these processes, we did not use this version of ParFlow in our study. We will add this discussion to our revised manuscript. |

**References:**

Abatzoglou, J. T. (2013). Development of gridded surface meteorological data for ecological applications and modelling. *International Journal of Climatology*, *33*(1), 121–131. https://doi.org/10.1002/joc.3413

Kollet, S.J., Cvijanovic, I., Schüttemeyer, D., Maxwell, R.M., Moene, A.F. and Bayer, P. (2009), The Influence of Rain Sensible Heat and Subsurface Energy Transport on the Energy Balance at the Land Surface. *Vadose Zone Journal*, 8: 846-857. https://doi.org/10.2136/vzj2009.0005

Pfister, L., Martínez-Carreras, N., Hissler, C., Klaus, J., Carrer, G. E., Stewart, M. K., and McDonnell, J. J. (2017). Bedrock geology controls on catchment storage, mixing, and release: A comparative analysis of 16 nested catchments. *Hydrological Processes*, 31(10), 1828–1845. https://doi.org/10.1002/hyp.11134

---

## Author Comment (AC2)

**Response to Reviewer #2:**

First of all, we would like to thank reviewer #2 for his/her comments on the paper. Their effort has helped us to improve the manuscript and we appreciate you agreeing to review the paper during these challenging times. Here, we provide point-by-point responses to each of reviewer 2's comments.

| Reviewer 2 comments | Author response |
|---|---|
| What could be added is a discussion of how the model parameterisation affects the conclusions. Such a discussion is started on page 14 but could be more comprehensive. | We agree with the reviewer comment here, and reviewer #1 had similar comments. In our previous work (Schreiner-McGraw and Ajami, 2020), we performed a limited set of simulations to test the impact of saprolite layer parameterization, the most hydrologically active zone in the subsurface, on simulated water budget. The parameterization of this geologic layer did not significantly impact the propagation of uncertainty in precipitation into the groundwater. Please see figure 11 in Schreiner-McGraw and Ajami, 2020. |
| | Our previous experiments were limited in scope by the high computational demands of running ParFlow.CLM. Unfortunately, that limitation applies to this current experiment as well and has prevented us from being able to use parameter uncertainty approaches such as the Generalized Likelihood Uncertainty Estimation (GLUE) to evaluate the full impact of model parameterization on our results. |
| | Model parameterization and geologic setting, however, likely play a role in how uncertainty in meteorological forcings will propagate into groundwater. In the revised manuscript, we will expand upon this discussion. For example, our simulations are performed in a mountain region underlain by fractured, low permeability bedrock. Previous work has shown that groundwater in these regions responds quickly to changes in precipitation (Pfister et al., 2017), and would likely impact the results. In the revised manuscript we will |

| | include discussion of how the geologic setting of our study site, and parameterization choices that we made are likely to impact the study results. |
|---|---|
| In terms of presentation, although the paper is generally well written, it is repetitive in places and the flow of arguments could be sharpened. | Thank you for the reminder. We will aim to improve the communication in the revised manuscript. We will improve our topic sentences for paragraphs to highlight the purpose of each discussion, and help with the flow of arguments. Finally, we will search for repetitive sentences and remove text to make the manuscript more concise. |

**References:**

Pfister, L., Martínez-Carreras, N., Hissler, C., Klaus, J., Carrer, G. E., Stewart, M. K., and McDonnell, J. J. (2017). Bedrock geology controls on catchment storage, mixing, and release: A comparative analysis of 16 nested catchments. *Hydrological Processes*, 31(10), 1828–1845. https://doi.org/10.1002/hyp.11134

Schreiner-McGraw, A.P. and Ajami, H. 2020. Impact of uncertainty in precipitation datasets on the hydrologic budget of an integrated hydrologic model in mountainous terrain. *Water Resources Research*, 56(12), doi: 10.1029/2020WR027639

---

## Author Response (AR1)

**Response to Reviewer #1:**

First of all, we would like to thank reviewer #1 for his/her comments on the paper. Their effort has helped us to improve the manuscript and we appreciate you agreeing to review the paper during these challenging times. Here, we provide point-by-point responses to each of reviewer 1's comments.

| Reviewer 1 comments | Author response |
| --- | --- |
| It needs to be demonstrated how/if the results of this study depend on the single year of the simulation. Why not extend the study period to include the last 30 years or so? All of the forcings used in this study go back to early 1980s. Even MODIS goes back to the early 2000s. A longer period of analysis can also help address the question of impact of forcing uncertainties on the long-term changes in mountain system recharge, which would be of interest given the focus on global warming driven changes in precipitation and temperature. A longer analysis period could also allow for independent verification of the mountain system recharge simulations such as by using GRACE based estimates of recharge, which goes back to early 2000s. This could help identify the set of atmospheric forcings which yield the most realistic estimates of the recharge. | You are correct, thank you. The model requires hourly forcing data and the spatially downscaled, hourly forcing that we use (Princeton CONUS Forcing) is only available from 2002, and other forcing data such as the original NLDAS-2 would be too coarse for this analysis. During the 2000-2019 period, California suffered from the worst drought in recorded history. Our initial thoughts were that if we used this period, it would likely bias the model sensitivity results based on these extreme years. But, as you state, it would make the findings more robust.

To address reviewer comment in the revised manuscript, we have added two more years of simulation in addition to the water year (WY) 2016 originally selected. Please note that (WY) 2016 approximately represents average precipitation and air temperature in the watershed. To assess the impact of hydroclimatic condition on our results, we performed simulations for WY2014 and WY2011, representing extreme dry and wet years in the catchment, respectively. Similar to our approach for WY2016, we evaluated model simulation results at equilibrium to remove the impact of initial condition bias on sensitivity analysis results. Integrated hydrologic models like ParFlow.CLM model are sensitive to initial conditions particularly for subsurface storages due to dynamic interactions between the subsurface and land surface fluxes. As a result, we chose to analyze simulations that had reached dynamic equilibrium conditions with respect to subsurface storages for each water year. It is |

| | not feasible to allow a 15-30 year simulation reach equilibrium because of the computational demands of the model. Therefore, we perform simulations for the select dry and wet water years at equilibrium conditions to assess generality of our results. |
|---|---|
| | While using GRACE data is ideal for confirming changes in terrestrial water storages, the resolution of GRACE data is too coarse for the study basin. Furthermore, we do not include irrigation and water management options in this version of the model so it is not possible to assess the impacts of forcing uncertainty over the entire basin. |
| Additionally, how are the results of this study dependent on the choice of the hydrologic model? As shown by Vano et al, 2012 (cited by this manuscript too) depending on the choice of hydrologic model sensitivity of hydrologic variables (such as runoff) to changes in precipitation and temperature can vary substantially. | We agree that the choice of hydrologic model will impact our results. In the revised manuscript, we made it clear that our results are applicable to integrated surface water – groundwater models that implement the 3D Richard's equation to simulate variably saturated subsurface flow across the entire subsurface, and have a fully integrated overland flow simulator. Of course, different model physics will result in different sensitivities. However, we are using the most physically-based approach for simulating surface water-groundwater processes, and have done detailed model validation to make sure major hydrologic processes are captured by the model. Ideally, one should perform such simulations using different model structures to assess the impact of all uncertainty sources on simulated hydrologic response. However, such model intercomparison beyond the scope of this study.

We have added a new section to the discussion, section 3.7, to discuss the impact of the hydrologic model and model parameterization on our results. On L571 we state: |

"In this study, we use a physically-based, integrated hydrologic model, ParFlow.CLM, to quantify the impact of uncertainty in meteorological forcings on the simulated groundwater. However, the generality of results are influenced by multiple factors as described below. The results from this study are applicable to integrated surface water – groundwater models that implement the 3D Richard's equation to simulate variably saturated subsurface flow across the entire subsurface, and have a fully integrated overland flow simulator. Previous studies have found that hydrologic sensitivities of land surface models can vary widely based on the model used (Vano et al., 2012).

We also explain how model parameterization might affect our simulated mountain system recharge. On L580 we state:

"Additionally, model parameterization is expected to affect the uncertainty to meteorological forcings. Previous results showed that three different conceptual models of the saprolite layer did not systematically impact the simulated groundwater response to precipitation variability (Schreiner-McGraw and Ajami, 2020). However, the simulated MFR depends on the subsurface permeability values assigned to the Central Valley aquifer in the piedmont slope region. Our hydraulic parameter values are based on drill core data and a previously calibrated hydrologic model (Faunt, 2009), but hydraulic conductivity values may be too high causing overestimation of simulated MFR (Brush et al., 2013). Historical observations under pre-development conditions suggest that the Kaweah River branched into several smaller distributaries, some of which did not flow all the way to the historic Tulare Lake (U.S. EPA, 2007; Hall, 1886). These observations suggest that our MFR estimates from the Kaweah River are reasonable, but are likely overestimated due to coarse horizontal model

| | resolution resulting in streambeds that are unreasonably wide and potentially overestimated hydraulic conductivity of the Central Valley sediments. Conversely, the coarse resolution of the model may result in an underestimation of MFR via small channels and first-order watersheds located on the piedmont slope (Schreiner-McGraw and Vivoni, 2018)." |
|---|---|
| Finally, it also should be at least discussed how the results of this study may depend on the choice of the study domain. | Yes, this is a good point. Model parameterization and geologic setting are likely play a major role in how uncertainty in meteorological forcings will propagate into groundwater. In the revised manuscript, we expanded upon this discussion. For example, our simulations are performed in a mountain region underlain by fractured, low permeability bedrock. Previous work has shown that groundwater in these regions responds quickly to changes in precipitation (Pfister et al., 2017), which would likely impact the results. While the role of uncertainty in precipitation forcing is discussed extensively, our main goal here was to highlight the role of temperature in addition to precipitation for regions with high relief. Of course, the obtained sensitivities in different mountain settings are impacted by the quality of meteorological forcings, topography, vegetation and subsurface characteristics.

On L592 of the revised manuscript we state:

"In addition to the hydrologic model structure, the selection of study domain will affect our results, and we would expect different sensitivities depending on the topography and vegetation type in other regions. Despite site specific nature of our study, the evergreen forest in our study watershed is broadly representative of evergreen forests in the mountainous, Western United States. In our simulations, the weathered bedrock zone is the most hydrologically active region of the |

| | subsurface, which has been observed as a feature of the Sierra Nevada (Holbrook et al., 2014). This is a common pattern in other mountainous regions with low-permeability bedrock (Jencso et al., 2009; Pfister et al., 2017; Spencer et al., 2019). Previous work in mountain regions with low-permeability bedrock has found that storage can respond quickly to meteorological conditions as a result of the low-permeability and low storage capacity (Pfister et al., 2017), and would impact the overall hydrologic response. Further research to examine how meteorological forcing uncertainty propagates into groundwater systems across a range of bedrock conditions is warranted." |
|---|---|
| I am surprised a bit about the differences in the simulated variables generated using GridMET, NLDAS and PRISM datasets. As described in Abatzoglou 2013, GridMET is based on the NLDAS-2 and PRISM dataset. Please at least discuss why this might be the case. | We agree that it is worth further highlighting these differences in the paper. We believe that the differences among products are caused by the fact that we are using different versions of PRISM and NLDAS-2 than the version used in Abatzoglou (2013) paper to generate the Gridmet dataset. To build the Gridmet dataset, they used the 800 m resolution version of PRISM, while we used the freely available 4 km resolution of PRISM data. Additionally, we used a downscaled version of the NLDAS-2 dataset, called the Princeton CONUS Forcing dataset with ~3 km resolution. As described in the paper, the Princeton dataset is the downscaled version of the original NLDAS-2 data with ~12 km resolution and the rainfall data is updated by using the radar products. We believe that the differences in the resolution of the dataset and interpolation approach have caused the differences in precipitation forcing datasets.

In the revised manuscript we state on L294:

"Differences in mean annual daily temperature from the mean temperature dataset range between -8 to 8 ⁰C (Fig. 2b-f). Differences in mean daily temperature among different forcing datasets exist irrespective of the wetness condition (wet vs dry or average |

| | year), and the ranges are larger for WY2011 (Fig. 3 a,c,e). Considerable uncertainty exists in the daily and annual totals of precipitation from the different gridded datasets as well (Fig. 3 b,d,f). The differences between the gridded products in our study are surprising, especially considering that Gridmet is based on the NLDAS-2 and PRISM datasets. We believe that the differences among products are caused by contrasting spatial resolutions. For example, Abatzoglou (2013) used the 800 m PRISM data to generate the Gridmet dataset, while we used the freely available PRISM data at 4 km resolution. Additionally, we used a downscaled version of the NLDAS-2 dataset, called the Princeton CONUS Forcing dataset at ~3 km resolution with the updated precipitation data using the Stage IV and Stage II radar products. We believe that the differences in the resolution of the datasets and interpolation approaches have caused the differences in precipitation and air temperature forcing datasets." |
|---|---|
| Page 1, line 28: "high qualify" should be "high quality" | Thank you for pointing this out, we have fixed this typo. |
| Page 8, 217-219, how does this chosen threshold of 2.5 deg C for partition of precipitation into rainfall and snow, affect the results of this analysis, especially in the mid to low elevation parts of the domain? | To be clear, this threshold was not chosen by us, it is the threshold that the CLM model uses to partition precipitation into rainfall and snow. That being said, this threshold likely impacts the results. However, we did not assess its impacts.

In the mid to low elevation portions of the domain, where precipitation can currently fall as either rain or snow, the snow melts quickly and snowpack does not accumulate due to higher temperatures in mid-elevation regions.

On L576 of the revised manuscript we state:

"The land surface model we employ, CLM, applies a threshold temperature of 2.5 °C, below which precipitation falls as snow, which could have implications for our results. However, we expect its impact to be minimal, as most of the snow falls when the air |

| | temperature is much less than 2.5 °C. Models with different rain/snow partitioning schemes, however, might find different sensitivities than what we describe here." |
|---|---|
| Page 22, lines 496-498, I am not sure why land surface temperature and soil moisture would not be affected by the choice of forcings, wouldn't changes in ET affect both? Please clarify. | Yes, changes in ET does affect both land surface temperature and soil moisture. At the annual scale, however, changes in soil moisture are small because changes in ET can be balanced out by changes in potential recharge and lateral soil moisture redistribution. In the revised manuscript, we clarified the text in this section.

We believe that lower sensitivity to land surface temperature is partly related to the simplification made to represent the ground heat flux calculation in CLM. Many land surface models, including CLM, only incorporate heat transport via conduction and this simplification decouples heat transport from soil moisture transport. Including heat convective transport through soil moisture distribution will increase computational time. While the ParFlowE model (Kollet et al., 2009) incorporates these processes, we did not use this version of ParFlow in our study. In the revised manuscript we state on L528:

"At the annual scale, the result of Θ is not surprising because the soil moisture is controlled by both ET and R, where an increase in one can be compensated by a decrease in the other flux. Additionally, variability of these fluxes is highest at a daily compared to the annual scale. We believe that lower sensitivity of $T_g$ is related to the simplification made to represent the ground heat flux calculation in CLM. To reduce computational time, many land surface models, including CLM, only incorporate heat transport via conduction and this simplification decouples heat transport from soil moisture transport (Kollet et al., 2009)." |

**References:**

Abatzoglou, J. T. (2013). Development of gridded surface meteorological data for ecological

applications and modelling. *International Journal of Climatology*, *33*(1), 121–131. https://doi.org/10.1002/joc.3413

Kollet, S.J., Cvijanovic, I., Schüttemeyer, D., Maxwell, R.M., Moene, A.F. and Bayer, P. (2009), The Influence of Rain Sensible Heat and Subsurface Energy Transport on the Energy Balance at the Land Surface. *Vadose Zone Journal*, 8: 846-857. https://doi.org/10.2136/vzj2009.0005

Pfister, L., Martínez-Carreras, N., Hissler, C., Klaus, J., Carrer, G. E., Stewart, M. K., and McDonnell, J. J. (2017). Bedrock geology controls on catchment storage, mixing, and release: A comparative analysis of 16 nested catchments. *Hydrological Processes*, 31(10), 1828–1845. https://doi.org/10.1002/hyp.11134

**Response to Reviewer #2:**

First of all, we would like to thank reviewer #2 for his/her comments on the paper. Their effort has helped us to improve the manuscript and we appreciate you agreeing to review the paper during these challenging times. Here, we provide point-by-point responses to each of reviewer 2's comments.

| Reviewer 2 comments | Author response |
|---|---|
| What could be added is a discussion of how the model parameterisation affects the conclusions. Such a discussion is started on page 14 but could be more comprehensive. | We agree with the reviewer comment here, and reviewer #1 had similar comments. In our previous work (Schreiner-McGraw and Ajami, 2020), we performed a limited set of simulations to test the impact of saprolite layer parameterization, the most hydrologically active zone in the subsurface, on simulated water budget. The parameterization of this geologic layer did not systematically impact the propagation of uncertainty in precipitation into the groundwater. Please see figure 12 in Schreiner-McGraw and Ajami, 2020.

Our previous experiments were limited in scope by the high computational demands of running ParFlow.CLM. Unfortunately, that limitation applies to this current experiment as well and has prevented us from being able to use parameter uncertainty approaches such as the Generalized Likelihood Uncertainty Estimation (GLUE) to evaluate the full impact of model parameterization on our results.

Model parameterization and geologic setting, however, likely play a role in how uncertainty in meteorological forcings will propagate into groundwater. In the revised manuscript, we will expand upon this discussion. For example, our simulations are performed in a mountain region underlain by fractured, low permeability bedrock. Previous work has shown that groundwater in these regions responds quickly to changes in precipitation (Pfister et al., 2017), and would likely impact the results. |

| | In the revised manuscript we have included a discussion section (section 3.7) to discuss the role of model domain, model selection, and model parameterization on our results. |
|---|---|
| In terms of presentation, although the paper is generally well written, it is repetitive in places and the flow of arguments could be sharpened. | Thank you for the reminder. We will aim to improve the communication in the revised manuscript. We have improved our topic sentences for paragraphs to highlight the purpose of each discussion, and help with the flow of arguments. We have also removed several repetitive sentences. Finally, we removed what was Figure 11, and its associated discussion because we did not believe they were essential to our findings. |

**References:**

Pfister, L., Martínez-Carreras, N., Hissler, C., Klaus, J., Carrer, G. E., Stewart, M. K., and McDonnell, J. J. (2017). Bedrock geology controls on catchment storage, mixing, and release: A comparative analysis of 16 nested catchments. *Hydrological Processes*, 31(10), 1828–1845. https://doi.org/10.1002/hyp.11134

Schreiner-McGraw, A.P. and Ajami, H. 2020. Impact of uncertainty in precipitation datasets on the hydrologic budget of an integrated hydrologic model in mountainous terrain. *Water Resources Research*, 56(12), doi: 10.1029/2020WR027639

---

## Author Response (AR2)

**Response to Reviewer #2:**

Thank you for reviewing the paper for a second time! Please see our response to your final comment below.

| Reviewer 2 comments | Author response |
|---|---|
| Thank you for providing your response to my comments. In general I find them to be satisfactory. I do however have a remaining and important comment. I am still confused about which dataset you mean by "Princeton CONUS forcings" dataset. Are you referring to MSWEP V2.2? If yes, then that dataset does go back to 1979, plus its a global dataset hence now mainly based on NLDAS-2, which is only available over North America. I am also confused because currently the Beck et al., 2019 paper that you are citing does not describe a new precipitation dataset it whereas compares several precipitation datasets including MSWEP. Do you mean to cite other Beck et al, 2019 paper which was published in BAMS?

 Also, I might suggest to use a different term than "Princeton CONUS forcings" for the forcings dataset you are using, as there are other forcings datasets generated by the Princeton Group, most notably Sheffield et al., 2006 dataset. | We are not referring to the MSWEP dataset. The dataset that we use is described in the citation to Pan et al. (2016), while the Beck et al. (2019) paper that we cite provides some validation and examination of the strengths and weaknesses of the dataset. We have clarified this by changing the text to read:

 "We obtain all meteorological forcings, except precipitation ($P$) and air temperature ($TA$), from the Princeton CONUS Forcing dataset, which provides hourly forcings at 3-km spatial resolution based on the NLDAS-2 dataset (Pan et al., 2016). This dataset downscales the NLDAS-2 precipitation dataset using Stage IV and Stage II radar products (Pan et al., 2016) and has been validated and compared to several other gridded datasets, showing good performance (Beck et al., 2019)."

 Regarding the name of the dataset, when we spoke with Dr. Ming Pan, who provided us access to the data, he instructed us to call it the "Princeton CONUS Forcings" dataset. So, we believe it is best to follow his preference. |